# DNA: Uncovering Universal Latent Forgery Knowledge

**Jingtong Dou** [* 1]  **Chuancheng Shi** [* 1 2]  **Anqi Yi** [* 1]  **Shiming Guo** [1]  **Wenhua Wu** [1]
**Yemin Wang** [3]  **Li Zhang** [4]  **Fei Shen** [† 2]  **Tat-Seng Chua** [2]

## Abstract

As generative AI achieves hyper-realism, superficial artifact detection has become obsolete. While prevailing methods rely on resource-intensive fine-tuning of black-box backbones, we propose that forgery detection capability is already encoded within pre-trained models rather than requiring end-to-end retraining. To elicit this intrinsic capability, we propose the discriminative neural anchors (DNA) framework, which employs a coarse-to-fine excavation mechanism. First, by analyzing feature decoupling and attention distribution shifts, we pinpoint critical intermediate layers where the focus of the model logically transitions from global semantics to local anomalies. Subsequently, we introduce a triadic fusion scoring metric paired with a curvature-truncation strategy to strip away semantic redundancy, precisely isolating the forgery-discriminative units (FDUs) inherently imprinted with sensitivity to forgery traces. Moreover, we introduce HIFI-Gen, a high-fidelity synthetic benchmark built upon the very latest models, to address the lag in existing datasets. Experiments demonstrate that by solely relying on these anchors, DNA achieves superior detection performance even under few-shot conditions. Furthermore, it exhibits remarkable robustness across diverse architectures and against unseen generative models, validating that waking up latent neurons is more effective than extensive fine-tuning.

## 1. Introduction

Modern generative models (Shen & Tang, 2024; Shen et al., 2024; 2025; Wu et al., 2025; Shen et al.; Zhang et al.,

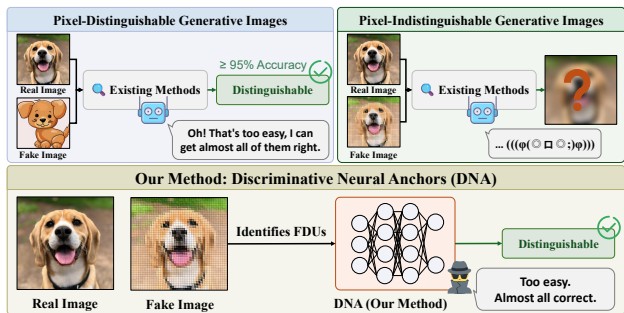

*Figure 1.* **Comparison of detection paradigms in the era of hyper-realistic generation.** Unlike conventional methods that fail as surface artifacts disappear, our DNA framework exploits neuronal hierarchies to robustly detect hyper-realistic scenarios.

2024a;b; 2025b;a) have achieved such hyper-realism that synthetic images now impeccably mimic the statistical patterns of natural images. This evolution renders traditional forensic methods, which rely on surface-level pixel or frequency artifacts, increasingly obsolete. As shown in Figure 1, when forgery becomes statistically indistinguishable from reality at the pixel level, superficial detectors inevitably lose efficacy. Consequently, there is an urgent need for authentication techniques that transcend surface appearances to explore the internal representation mechanisms of deep learning models.

To address this challenge, the prevailing defense paradigm primarily (Wang et al., 2020; Lei et al., 2026; Liu et al., 2026; Zhang et al., 2025c; Park et al., 2025; Wang et al., 2023; Ojha et al., 2023) adopts a strategy of "feature extraction followed by full fine-tuning". Researchers widely deploy large-scale pre-trained vision models (e.g., CLIP (Radford et al., 2021), ViT) to construct feature spaces; however, they often treat these models as mere "black boxes" for global feature extraction. This approach not only overlooks the intricate neural activation patterns within the models but also necessitates expensive full-parameter fine-tuning on massive forgery datasets to "acquire" detection capabilities. However, the substantial computational and data costs associated with this paradigm compel us to revisit a fundamental question: **Is the ability to detect forgeries truly a skill that must be acquired through extensive "post-hoc training" on massive data?** Or does an "intrinsic intuition" for distinguishing authenticity already lie dormant within

---

[*]Equal contribution [†]Corresponding author. [1]The University of Sydney, Sydney, Australia [2]National University of Singapore, Singapore [3]Xiamen University, Xiamen, China [4]The Hong Kong Polytechnic University, Hong Kong, China. Correspondence to: Fei Shen <shenfei29@nus.edu.sg>.

*Proceedings of the 43rd International Conference on Machine Learning*, Seoul, South Korea. PMLR 306, 2026. Copyright 2026 by the author(s).

the depths of powerful pre-trained representations, having long overlooked? This prompts a critical reflection: rather than laboriously instilling detection knowledge into models, is it possible to excavate the "instinct" for authenticity discrimination latent within these pre-trained representations?

This paper challenges the fine-tuning paradigm by proposing that forgery detection is not an acquired skill from posterior training, but latent knowledge inherent in the pre-trained backbone. We argue that massive pre-training on natural data inherently encodes sensitivity to generative artifacts within specific sparse neurons. While these neurons remain dormant during standard semantic tasks, they form the fundamental discriminative basis for authenticity. Consequently, our objective shifts from training the model to "learn" detection to designing a probing mechanism that elicits and extracts this off-the-shelf latent knowledge.

To precisely extract this latent knowledge, we propose the discriminative neural anchors (DNA) framework, featuring a resource-efficient "coarse-to-fine" excavation mechanism. At the coarse-grained level, we localize critical depth intervals by capturing the functional transition from global semantics to local artifacts. Subsequently, at the fine-grained level, we introduce a triadic fusion scoring mechanism that integrates gradient sensitivity, activation magnitude, and weight contribution to precisely isolate the forgery-discriminative units (FDUs). This sparse ensemble of neurons captures the intrinsic "DNA fingerprints" of forgery while stripping away semantic redundancy. Empirical results substantiate that by relying solely on this compact subset of FDUs, the model achieves superior detection performance under few-shot, demonstrating remarkable robustness across diverse architectures and unseen generative models. This finding not only validates the "less is more" principle but also reveals that unlocking the latent knowledge of pre-trained models is a promising new avenue for building generalized, efficient forgery-detection systems. Finally, to address the lag in existing benchmarks, we introduce HIFI-Gen, a high-fidelity dataset featuring the latest models, providing a more challenging evaluation for modern forgery detection. We highlight the following contributions:

- We uncover "sleeping genes" within pre-trained models, demonstrating that core discriminative evidence resides in sparse, long-tailed deep neurons.

- We propose the discriminative neural anchors (DNA) framework. It extracts a compact, sensitive subspace in a coarse-to-fine manner, achieving "less is more" efficiency.

- We introduce HIFI-Gen, a benchmark for high-fidelity synthesis. By covering multiple cutting-edge models, it establishes a robust foundation for future research in advanced generative logic.

## 2. Related Work

**AI-generated image detection.** The field of AI-generated image detection has evolved significantly to keep pace with the advancing capabilities of generative models. Early approaches primarily exploited frequency fingerprints or spatial artifacts (Wang et al., 2020; Chai et al., 2020); however, these explicit features are becoming obsolete as generation quality improves. Consequently, recent methods such as DIRE (Wang et al., 2023), AEROBLADE (Ricker et al., 2024) and LaRe2 (Luo et al., 2025) have shifted toward leveraging reconstruction errors from diffusion models as discriminative cues. Simultaneously, pre-trained models like CLIP have demonstrated great potential through either lightweight probing (Ojha et al., 2023; Cozzolino et al., 2024) or parameter-efficient fine-tuning (PEFT)(Liu et al., 2023; 2024; Zhou et al., 2025). Despite these advancements, existing approaches typically treat detection as a capability that must be "learned" through additional fine-tuning, often relying on black-box backbones. Even methods exploring intermediate features (Park et al., 2025) or feature decoupling (Zhang et al., 2025c; Liu et al., 2025a) still depend on external training paradigms. In contrast, we propose the discriminative neural anchors (DNA) framework. We posit that detection capability is an inherent latent knowledge within pre-trained models, enabling universal and intrinsic detection without the need for extensive fine-tuning.

**Neuron Interpretability.** Research on low-level perception and macro-alignment suggests that pre-trained models naturally encapsulate the understanding of essential image features through large-scale learning (Zhang et al., 2018; Shi et al., 2025; Schwettmann et al., 2023; Huang et al., 2025a). Simultaneously, advancements in mechanistic interpretability offer robust tools for extracting this latent knowledge. Specifically, hierarchical feature reconstruction via SAEs(Huang et al., 2025b; Zaigrajew et al., 2025) and the identification of domain-specific neurons (Huo et al., 2024; Lim et al., 2025) demonstrate that models contain highly specialized sparse representations. By incorporating neuron importance scoring (Xie et al., 2021; Hu et al., 2025; Tang et al., 2024; Chen et al., 2024b), we can precisely anchor sparse subsets sensitive to forgery traces. The intersection of these dimensions forms the foundation of our DNA framework. DNA marks a paradigm shift from "acquired trait" (learning via fine-tuning) to "latent knowledge" (awakening internal representations).

## 3. Construction of HIFI-Gen

To comprehensively evaluate the generalization capability of forgery detectors against state-of-the-art synthesis algorithms, we constructed a high-fidelity generation dataset named HIFI-Gen. The construction of HIFI-Gen followed a systematic pipeline: following the protocols established

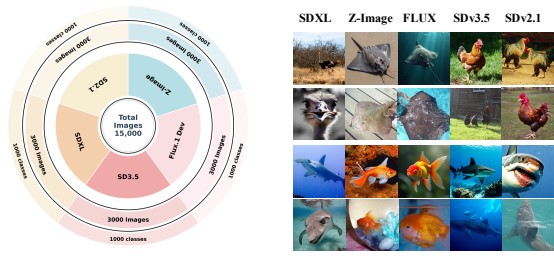

*Figure 2.* **Visualization of HIFI-Gen structure.** HIFI-Gen comprises images generated by five distinct generative models, each yielding 3,000 images across 1,000 categories.

by GenImage (Zhu et al., 2023), we utilized ImageNet class labels as semantic prompts to ensure both content diversity and semantic alignment. We employed five advanced text-to-image architectures representing distinct generative paradigms: SDv3.5(Esser et al., 2024), SDXL (Podell et al., 2023), and SDv2.1 (Rombach et al., 2022), the flow-matching-based Flux.1 Dev (Labs, 2024; Labs et al., 2025), and the proprietary Z-Image (Team et al., 2025; Liu et al., 2025b; Jiang et al., 2025). As illustrated in Figure 2, the final dataset comprises 15k images in total, with 3k samples synthesized by each generator across a broad range of categories to ensure balanced distribution. While existing benchmarks predominantly rely on legacy GAN or early U-Net architectures with discernible structural traces, HIFI-Gen distinguishes itself by strategically focusing on contemporary Diffusion Transformers (DiT) and Flow-matching technologies. By bridging the "architectural gap" left by obsolete datasets and simulating the "near-zero-artifact" challenge of next-generation synthetic content, HIFI-Gen ensures that our DNA framework is evaluated against the most photorealistic forgeries, providing a more rigorous assessment of generalization in real-world scenarios.

## 4. Method

As shown in Figure 3, to systematically localize the layers most sensitive to forgery signals, we design a hierarchical probing framework that operates in a coarse-to-fine manner. Instead of searching the entire parameter space, we first initiate a coarse-grained layer localization phase. By quantifying the discriminative efficacy of representations at each depth and analyzing statistical discrepancies in the feature space, we lock onto the intermediate layer intervals that are most sensitive to forgery signals, creating a precise search candidate pool. Building on this, we proceed to the fine-grained neuron screening phase, where we surgically isolate the sparse key neurons (i.e., FDUs) that capture intrinsic forgery traces from the localized layers.

### 4.1. Layer Localization

**Coarse-grained Layer Localization.** To efficiently pinpoint the proposed forgery-discriminative units (FDUs), we

first constrain the vast search space to specific depth intervals. We use cosine distance to measure the geometric separation between authentic and forged representations in feature space. For the $i$-th layer, let $f^{(i)}(x)$ denote the extracted $CLS$ token for an input image $x$. We compute the class centroids $\mu_{c,i}$ for each class $c \in \{real, fake\}$ as:

$$\mu_{c,i} = \frac{1}{|X_c|} \sum_{x \in X_c} f^{(i)}(x), \tag{1}$$

where $X_c$ denotes the set of images in class $c$. The discrepancy is then quantified as:

$$D_{cos}(i) = 1 - \frac{\mu_{real,i} \cdot \mu_{fake,i}}{||\mu_{real,i}||||\mu_{fake,i}||}. \tag{2}$$

As shown in Figure 4, in early layers, $D_{cos}(i) \approx 0$, indicating that features are entangled and highly similar in their directional orientation. However, a sharp rise in $D_{cos}(i)$ is observed starting from layer 8. This suggests that the 'neural fingerprints' of forgery exhibit a significant directional shift from authentic features within this depth interval, thereby maximizing linear separability.

Although cosine distance can verify the global separability of features, it fails to explain the spatial origin of such differences. To verify whether this feature decoupling is driven by the model focusing on specific visual anomalies, we further analyze the attention allocation logic. Let $A^{(i)}(x) \in \mathbb{R}^L$ denote the global attention vector of the $[CLS]$ token for image $x$ at layer $i$, averaged across all heads. Similar to Eq. (1), we compute the mean attention vector $\bar{A}_{c,i}$ for each class $c$:

$$\bar{A}c, i = \frac{1}{|X_c|} \sum_{x \in X_c} A^{(i)}(x). \tag{3}$$

The discrepancy in spatial focus is then quantified by the euclidean distance between these class-wise means:

$$D_{L2}(i) = ||\bar{A}_{real,i} - \bar{A}_{fake,i}||_2. \tag{4}$$

As illustrated in Figure 4, a sharp peak in $D_{L2}(i)$ emerges specifically within the intermediate layers (e.g., layers 12–18). This aligns with the feature-space findings and indicates a drastic shift in spatial logic.

**Linear Probing.** For each layer $i$, we extract the $[CLS]$ token $h_i = f^{(i)}(x)$ and attach an independent, learnable linear head parameterized by $W_i$ and $b_i$. The datasetication logit is computed as:

$$z_i = W_i^\top h_i + b_i. \tag{5}$$

We employ lightweight linear probing independently using the available training data. The optimization objective

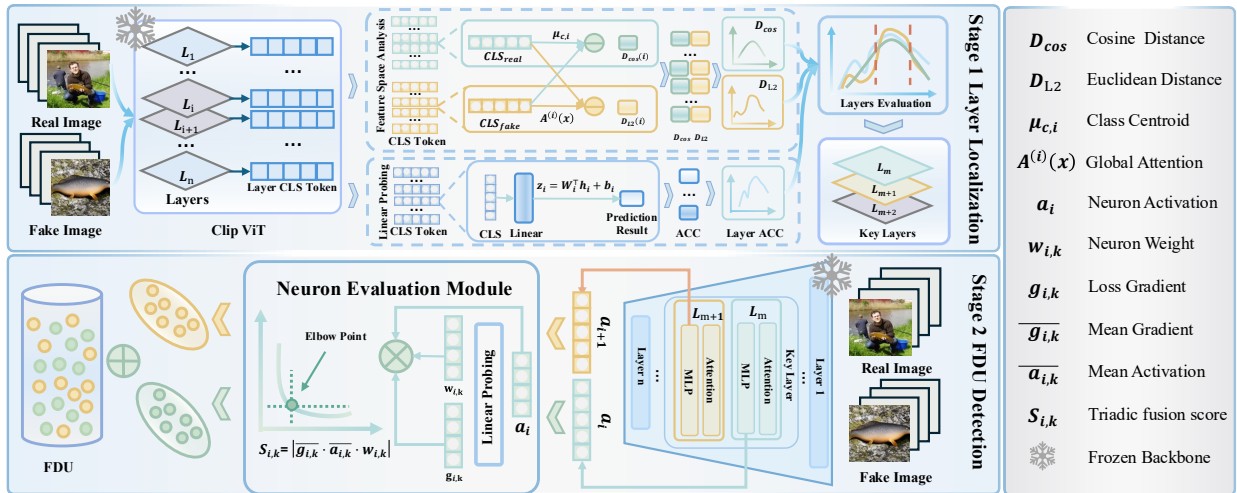

Figure 3. **Overall of the DNA framework.** The pipeline operates in a coarse-to-fine manner. Stage 1: layer localization. We pinpoint critical intermediate layers by analyzing feature-space decoupling and attention-distribution shifts, validated by linear probing. Stage 2: FDUs detection. We identify sparse forgery-discriminative units (FDUs) from the frozen backbone using a triadic fusion score ($S_{i,k}$).

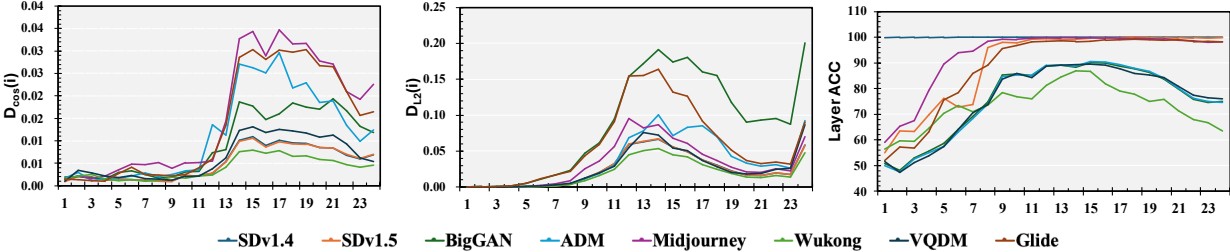

Figure 4. **Visualization of distance metrics across layers.** The cosine distance ($D_{cos}$) between the centroids of real and fake classes at each layer. The euclidean distance ($D_{L2}$) of the global attention distributions between real and fake images.

is to minimize the binary cross-entropy loss between the predicted probability $\sigma(z_i)$ and the ground truth label $y$:

$$\mathcal{L}i = -\mathbb{E}(x,y)\left[y\log(\sigma(z_i)) + (1-y)\log(1 - \sigma(z_i))\right], \quad (6)$$

where $\sigma(\cdot)$ denotes the sigmoid function. As shown in Figure 4, by evaluating the probing accuracy across different layers, we can pinpoint the depth interval where the latent representations possess the most potent and robust discriminative power for forgery detection.

### 4.2. Forgery-Discriminative Units Detection

Having localized the core search space to the critical layer interval, we proceed to the fine-grained extraction of FDUs to prune redundant dimensions and construct a compact forgery-sensitive subspace. To robustly evaluate each neuron's contribution, we propose a triadic fusion score ($S_{i,k}$) that integrates three complementary perspectives. Specifically, for the $k$-th neuron in layer $i$, we consider t: (i) The activation ($a_{i,k} \in \mathbb{R}$), which measures the magnitude of the neuron's response to the input; (ii) The weight ($w_{i,k} \in \mathbb{R}$), derived from the previously optimized linear probe, reflecting the neuron's statistical contribution to discriminability; (iii) The gradient ($g_{i,k} \in \mathbb{R}$), which quantifies the sensitivity

of the classification loss with respect to the neuron's activation. To ensure statistical stability, we compute the expected gradient magnitude $\bar{g}_{i,k}$ and the mean activation $\bar{a}_{i,k}$. We then combine these metrics to formulate the triadic fusion score $S_{i,k} \in \mathbb{R}$:

$$S_{i,k} = |\bar{g}_{i,k} \cdot \bar{a}_{i,k} \cdot w_{i,k}|, \quad (7)$$

prioritizing neurons that are simultaneously active, strongly weighted, and sensitive to forgery signals.

To automate FDUs selection without manual thresholding, we employ a global curvature-based truncation strategy. We first map the scores of all candidate neurons into a unified $[0, 1]$ scale using min-max normalization to eliminate magnitude discrepancies across depths. These normalized scores are sorted to form a ranking curve, upon which we apply the kneedle algorithm (Satopaa et al., 2011) to identify the "elbow point" $k^*$ defined as the index maximizing the perpendicular distance to the curve's start-end chord. Neurons ranked prior to $k^*$ are selected to form the FDUs signature, and their activations are concatenated into the final compact feature vector.

*Table 1.* **Cross-dataset evaluation on ForenSynths (Wang et al., 2020) and GenImage (Zhu et al., 2023) benchmarks.** We report ACC (%) and AP (%) scores to evaluate the generalization capability across various generative architectures. **Bold** and underline indicate the best and second-best performance, respectively.

| Dataset | Method | ProGAN | | StyleGAN | | StyleGAN2 | | BigGAN | | CycleGAN | | StarGAN | | GauGAN | | Deepfake | | Mean | |
|---|---|---|---|---|---|---|---|---|---|---|---|---|---|---|---|---|---|---|---|
| | | ACC | AP | ACC | AP | ACC | AP | ACC | AP | ACC | AP | ACC | AP | ACC | AP | ACC | AP | ACC | AP |
| ForenSynths | CNNDetection (Wang et al., 2020) | 91.4 | 99.4 | 63.8 | 91.4 | 76.4 | 97.5 | 52.9 | 73.3 | 72.7 | 88.6 | 63.8 | 90.8 | 63.9 | 92.2 | 51.7 | 62.3 | 67.1 | 86.9 |
| | PatchFor (Chai et al., 2020) | 97.8 | 100.0 | 82.6 | 93.1 | 83.6 | 98.5 | 64.7 | 69.5 | 74.5 | 87.2 | **100.0** | 100.0 | 57.2 | 55.4 | 85.0 | 93.2 | 80.7 | 87.1 |
| | LGrad (Tan et al., 2023b) | 99.9 | 100.0 | 94.8 | 99.9 | 96.0 | 99.9 | 82.9 | 90.7 | 85.3 | 94.0 | 99.6 | 100.0 | 72.4 | 79.3 | 58.0 | 67.9 | 86.1 | 91.5 |
| | UnivFD (Ojha et al., 2023) | 99.7 | 100.0 | 89.0 | 98.7 | 83.9 | 98.4 | 90.5 | 99.1 | 87.9 | 99.8 | 91.4 | 100.0 | 89.9 | 100.0 | 80.2 | 90.2 | 89.1 | 98.3 |
| | DIRE (Wang et al., 2023) | 98.3 | 99.9 | 72.5 | 94.3 | 66.3 | 97.7 | 59.1 | 79.2 | 66.2 | 78.2 | 92.8 | 100.0 | 54.9 | 72.2 | 87.0 | 97.1 | 74.6 | 89.8 |
| | NPR (Tan et al., 2023a) | 99.8 | 100.0 | 96.3 | 99.8 | 97.3 | 100.0 | 87.5 | 94.5 | 95.0 | 99.5 | 99.7 | 100.0 | 86.6 | 88.8 | 77.4 | 86.2 | 92.5 | 96.1 |
| | DRCT (Conv-B) (Chen et al., 2024a) | 98.6 | 99.9 | 76.2 | 96.2 | 60.8 | 96.2 | 86.0 | 75.7 | 96.9 | 98.0 | 61.4 | 94.1 | 82.6 | 99.5 | 33.9 | 75.3 | 74.6 | 91.9 |
| | DRCT (CLIP-L) (Chen et al., 2024a) | 98.4 | 99.9 | 82.8 | 95.0 | 74.5 | 95.4 | 87.9 | 96.7 | 92.2 | 98.0 | 83.2 | 95.8 | 98.5 | 99.9 | 42.4 | 78.6 | 82.5 | 94.9 |
| | MoLD (Park et al., 2025) | 99.9 | 100.0 | 91.1 | 99.8 | 86.0 | 99.8 | 98.4 | 100.0 | 98.1 | 99.9 | 99.1 | 100.0 | 99.7 | 100.0 | 58.7 | 96.5 | 91.4 | 99.5 |
| | **DNA (Ours)** | 99.9 | 100.0 | 96.2 | 99.9 | 97.9 | 99.9 | 95.2 | 99.2 | 99.9 | 99.9 | 99.7 | 100.0 | 99.6 | 99.9 | 89.2 | 97.2 | 97.2 | 99.5 |

| Dataset | Method | ADM | | BigGAN | | GLIDE | | Midjourney | | SDv1.4 | | SDv1.5 | | VQDM | | Wukong | | Mean | |
|---|---|---|---|---|---|---|---|---|---|---|---|---|---|---|---|---|---|---|---|
| | | ACC | AP | ACC | AP | ACC | AP | ACC | AP | ACC | AP | ACC | AP | ACC | AP | ACC | AP | ACC | AP |
| GenImage | CNNDetection (Wang et al., 2020) | 99.7 | 100.0 | 87.3 | 99.3 | 96.2 | 99.6 | 64.6 | 90.1 | 57.8 | 86.2 | 58.0 | 87.1 | 80.0 | 97.7 | 54.8 | 79.1 | 74.8 | 92.4 |
| | PatchFor (Chai et al., 2020) | 100.0 | 100.0 | 50.0 | 97.4 | 98.1 | 100.0 | 51.3 | 87.7 | 50.0 | 69.2 | 50.0 | 68.7 | 100.0 | 100.0 | 50.0 | 71.1 | 68.7 | 86.8 |
| | LGrad (Tan et al., 2023b) | 99.9 | 100.0 | 57.2 | 98.0 | 94.2 | 99.6 | 62.1 | 93.5 | 56.5 | 89.4 | 56.4 | 89.7 | 92.4 | 99.8 | 53.9 | 81.9 | 71.6 | 94.0 |
| | UnivFD (Ojha et al., 2023) | 90.6 | 97.1 | 91.7 | 99.1 | 79.0 | 88.2 | 61.8 | 69.5 | 80.3 | 89.2 | 79.5 | 88.4 | 90.8 | 98.5 | 84.8 | 93.3 | 82.3 | 90.4 |
| | DIRE (Wang et al., 2023) | 99.5 | 100.0 | 67.6 | 94.8 | 94.3 | 99.1 | 62.7 | 88.1 | 56.3 | 80.2 | 56.3 | 80.7 | 73.4 | 95.7 | 54.2 | 73.4 | 70.5 | 89.0 |
| | NPR (Tan et al., 2023a) | 100.0 | 100.0 | 51.2 | 98.6 | 96.1 | 99.8 | 64.4 | 90.9 | 52.9 | 77.9 | 52.7 | 78.6 | 74.9 | 97.8 | 52.3 | 73.8 | 63.5 | 88.2 |
| | DRCT (Conv-B) (Chen et al., 2024a) | 99.6 | 100.0 | 65.9 | 96.3 | 96.5 | 99.8 | 67.0 | 94.9 | 68.9 | 96.4 | 68.2 | 96.5 | 81.5 | 98.8 | 63.9 | 93.9 | 76.4 | 97.1 |
| | DRCT (CLIP-L) (Chen et al., 2024a) | 95.4 | 99.3 | 86.5 | 97.3 | 93.5 | 99.0 | 57.6 | 77.4 | 71.0 | 90.4 | 70.1 | 90.4 | 91.3 | 98.5 | 69.5 | 89.8 | 79.4 | 92.9 |
| | MoLD (Park et al., 2025) | 99.3 | 100.0 | 83.3 | 97.9 | 91.1 | 99.0 | 76.3 | 95.2 | 88.2 | 98.5 | 87.0 | 98.3 | 93.5 | 99.2 | 86.5 | 98.1 | 88.2 | 98.2 |
| | **DNA (Ours)** | 99.9 | 100.0 | 97.0 | 99.5 | 96.6 | 99.8 | 93.7 | 98.2 | 96.5 | 99.3 | 96.7 | 99.4 | 94.8 | 99.8 | 96.5 | 99.1 | 96.5 | 99.4 |

*Figure 5.* **Visualization of FDUs attention.** The activation maps of FDUs on both original and augmented images.

# 5. Experiments And Analysis

## 5.1. Implementation Details

**Dataset.** Following MoLD (Park et al., 2025), our experiments are primarily conducted on the GenImage (Zhu et al., 2023) and ForenSynths datasets (Wang et al., 2020). To evaluate the generalization performance against the latest text-to-image (T2I) methods, we constructed a test-only dataset HIFI-Gen. This dataset comprises images generated by novel models, including SDv3.5 (Esser et al., 2024), SDXL (Podell et al., 2023), SDv2.1 (Rombach et al., 2022), Flux.1 Dev (Labs, 2024; Labs et al., 2025) and Z-Image (Team et al., 2025; Liu et al., 2025b; Jiang et al., 2025) are aiming to assess the models' robustness against unseen generation logics.

**Metrics.** Following MoLD (Park et al., 2025), we adopt ACC and AP as core metrics to comprehensively quantify defensive performance across heterogeneous benchmarks and multi-source generation mechanisms. Additionally, we report the equal error rate (EER) to evaluate the trade-off between false acceptance and false rejection rates.

**Hyperparameters.** To demonstrate the effectiveness of the proposed hierarchical aggregation strategy and the key neuron extraction algorithm, this study adopts the core experimental settings from MoLD (Park et al., 2025). All core hyperparameters, including the base learning rate and decay strategies, are directly inherited from the MoLD configuration to ensure a fair comparison. All probing experiments and subspace training tasks were completed on a single NVIDIA GeForce RTX 3090 GPU.

*Table 2.* **Evaluation on HIFI-Gen dataset.** We evaluate the adaptability of detection methods to unseen architectures using our proposed HIFI-Gen benchmark. The table reports ACC and AP across five SOTA generators that were not included in the training stage.

| Method | SDv2.1 ACC | SDv2.1 AP | SDv3.5 ACC | SDv3.5 AP | SDXL ACC | SDXL AP | FLUX ACC | FLUX AP | Z-Image ACC | Z-Image AP | Mean ACC | Mean AP |
|---|---|---|---|---|---|---|---|---|---|---|---|---|
| CNNDetection (Wang et al., 2020) | 50.0 | 54.2 | 51.5 | 62.1 | 51.3 | 65.0 | 49.8 | 43.8 | 50.6 | 59.4 | 50.6 | 56.9 |
| PatchFor (Chai et al., 2020) | 60.2 | 63.2 | 63.1 | 68.9 | 65.4 | 73.0 | 70.7 | 80.0 | 60.3 | 68.3 | 63.9 | 70.7 |
| UnivFD (Ojha et al., 2023) | 73.4 | 77.7 | 67.1 | 67.9 | 71.7 | 76.5 | 52.0 | 48.6 | 63.4 | 60.9 | 65.5 | 66.3 |
| DIRE (Wang et al., 2023) | 55.8 | 63.1 | 57.2 | 72.4 | 55.4 | 61.1 | 53.3 | 57.4 | 54.8 | 57.9 | 55.3 | 62.4 |
| DRCT (Conv-B) (Chen et al., 2024a) | **97.8** | **99.8** | 88.7 | 99.1 | **95.7** | **99.7** | 87.8 | 98.2 | 85.5 | 99.4 | 91.1 | 99.2 |
| DRCT (CLIP-L) (Chen et al., 2024a) | 87.5 | 95.6 | 82.5 | 91.5 | 84.8 | 93.7 | 83.0 | 92.5 | **94.2** | 99.6 | 86.4 | 94.6 |
| MoLD (Park et al., 2025) | 82.8 | 99.4 | 88.1 | **99.5** | 74.3 | 98.8 | 89.5 | 99.7 | 70.4 | 98.1 | 81.0 | 99.1 |
| **DNA (Ours)** | 96.2 | 99.4 | **94.1** | 99.1 | 93.7 | 98.9 | **95.0** | **99.8** | 91.6 | **99.6** | **96.4** | 99.2 |

## 5.2. Compare With SOTA Methods

**Quantitative Result.** To verify the universality and robustness of our method across diverse generative mechanisms, we conducted cross-dataset evaluations on both the GAN-based ForenSynths (Wang et al., 2020) and diffusion-based GenImage (Zhu et al., 2023) benchmarks. The results indicate that our FDUs method consistently outperforms state-of-the-art competitors. Specifically, as shown in Table 1, our method attains a mean accuracy of 97.2% on ForenSynths and 96.5% on GenImage, surpassing the strongest baseline (MoLD (Park et al., 2025) ) by margins of 5.8% and 8.3%, respectively; notably, on the challenging Midjourney subset, we achieve 93.7% accuracy compared to MoLD's 76.3%, delivering a substantial 17.4% improvement. These findings demonstrate that the FDUs we have unearthed serve as a highly sensitive discriminator, capable of identifying subtle forgery traces with unparalleled precision even within the most intricate generative scenarios. They further confirm that our approach achieves stable and indeed superior performance with minimal resource expenditure, validating that excavating latent neural patterns is more effective than resource-intensive model fine-tuning.

**Qualitative Result.** To visually validate how the DNA framework captures the model's internal discriminative logic, we conducted an attention visualisation analysis on the FDUs. As shown in Figure 5, the attention of the FDUs precisely locks onto localised regions within generated images that exhibit logical inconsistencies, unnatural textures, or rendering defects. Conversely, for real images, their activation levels are significantly suppressed. This highly focused "neural probe" behaviour effectively circumvents interference from global semantics and background noise, demonstrating that FDUs do not rely on simple colour biases or statistical shortcuts. Instead, they genuinely penetrate the surface to capture underlying flaws in the pixel construction of the generative algorithm. Thus, the visualisation results robustly support the "discriminative awakening" hypothesis: by localising these sparse, highly specialised neurons, DNA achieves precise detection of forgery traces within a large-scale pre-training framework.

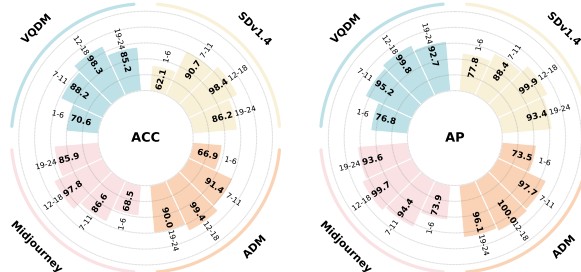

*Figure 6.* **Ablation study on layer depth intervals.** To analyze detection capabilities by depth, we partitioned the ViT into four segments. The following bar charts compare their ACC and AP scores across four diverse datasets.

**Generalization on Unseen Architectures.** To evaluate the few-shot adaptability of detectors to emerging generative paradigms, we conducted evaluations on the HIFI-Gen dataset, which includes the latest diffusion models (e.g., FLUX (Labs, 2024; Labs et al., 2025), SDv3.5 (Esser et al., 2024)), without any additional fine-tuning. As shown in Table 2, traditional methods like CNNDetection (Wang et al., 2020) and DIRE (Wang et al., 2023) fail to generalize, exhibiting near-random performance (50-55%). While recent methods like MoLD (Park et al., 2025) show improved detection on specific subsets, they suffer from significant volatility, dropping to 70.4% on Z-Image (Team et al., 2025; Liu et al., 2025b; Jiang et al., 2025). In contrast, our DNA framework achieves the highest Mean Accuracy of 96.4%, surpassing the strong competitor DRCT (Conv-B) (Chen et al., 2024a) by 5.3% and MoLD by 15.4%. Crucially, on the most advanced architectures such as FLUX and SDv3.5, our method outperforms DRCT(Conv-B) by margins of 7.2% and 5.4%, respectively. These results show that our approach remains robust to rapidly evolving generative models by leveraging intrinsic neuron-level discrimination rather than superficial statistics.

## 5.3. Ablation Study

**Layers Validation.** To validate the hypothesis that forgery-discriminative knowledge is concentrated within specific depth intervals, we conduct extensive ablation studies on different layer clusters of the pre-trained backbone. We

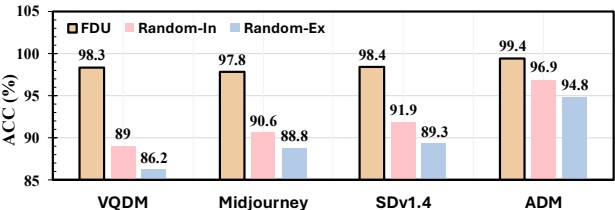

*Figure 7.* **Validation of neuron specificity.** We compare the detection accuracy of the proposed FDUs against two random selection baselines: Random-In (randomly selecting neurons within the same layer) and Random-Ex (randomly selecting neurons while explicitly excluding the identified FDUs).

*Table 3.* **FDUs validation.** We evaluate the importance of the identified neurons by measuring the performance drop when they are deactivated. **The lower the value, the greater the impact.**

| Method | ACC (↑) | AP (↑) |
|---|---|---|
| Clip-VIT-Large | 93.1 | 97.9 |
| + Masked Random Neurons | 91.1 (-2.0) | 97.1 (-0.8) |
| + Masked Hard Random | 92.0 (-1.1) | 97.7 (-0.2) |
| **+ Masked FDUs** | **65.1** (-28.0) | **90.6** (-7.3) |
| Clip-VIT-Large (Midjourney) | 81.6 | 91.4 |
| + Masked Random Neurons | 81.1 (-0.5) | 90.6 (-0.8) |
| + Masked Hard Random | 81.2 (-0.4) | 90.8 (-0.6) |
| **+ Masked FDUs** | **59.3** (-22.3) | **83.6** (-7.8) |

partition the 24-layer ViT model into four distinct blocks: Shallow (layers 1-6), Transition (layers 7-11), Critical (layers 12-18), and Deep (layers 19-24). All variants are tested under a few-shot setting with a training size of 500. As shown in Figure 6, the selected critical layers exhibit a decisive performance leap. On SDv1.4 model, ACC rises from 62.1% (Shallow) and 90.7% (Transition) to 98.4% (Critical), before declining in the deep layers to 86.2%. AP follows a similar trend, peaking at 99.9%. The experimental results provide strong evidence that the FDUs' core resides within the intermediate layer range. This confirms the precision of the hierarchical localization in the DNA framework: universal detection is "awakened" only by bypassing low-level noise and high-level semantic interference.

**Neurons Validation.** To further validate the specificity of FDUs and exclude the contribution of global representations, we conducted a comparative experiment across three groups: FDUs (ours), Random-In (randomly selecting equal neurons within the same layer), and Random-Ex (randomly selecting neurons while explicitly excluding FDUs). Results indicate that forgery-discriminative knowledge is not distributed globally but is strictly anchored to the sparse set of specialized neurons identified by DNA. Specifically, on the SDv1.4 model, excluding FDUs (Random-Ex) results in a precipitous decline in accuracy from 98.4% to 89.25%. Furthermore, as shown in Figure 7, the FDUs variant consistently outperforms the Random-In group across diverse models. Therefore, these findings establish the irreplaceability of FDUs in authenticity discrimination. By eliminating semantic redundancy, the DNA framework successfully "awakens" the intrinsic, cross-model universal knowledge latent in the pre-trained backbone.

**Functional Specificity & Robustness.** Table 3 provides converging evidence that FDUs form a functionally specific and stable subspace for forgery detection. In the base model, masking the identified FDUs reduces ACC from 93.1% to 65.1% (-28.0%). This drop is not explained by indiscriminate neuron removal or numerical magnitude: random masking and hard random masking (magnitude-matched non-FDUs) yield substantially smaller degradations, ruling out the hypothesis that FDUs matter merely because they

have large activations. Moreover, the identified FDUs are highly reproducible across independent trials with different random seeds and training subset samplings, exhibiting low variance and high overlap; we therefore report mean performance across trials. Finally, On the unseen Midjourney domain, masking the same pre-defined FDU indices still causes a pronounced collapse (ACC -22.3%), demonstrating robust cross-domain universality and confirming that detection relies on a statistically stable, functionally focused, and generalizable low-dimensional neuronal subspace.

### 5.4. More Result

**Robustness to Train-Data Size.** As shown in Figure 8, to assess the DNA framework's robustness to training data size, we conducted a sensitivity test across scales from 200 to 30,000 images. Experimental results reveal a distinct performance knee at approximately 5,000 samples, where marginal gains vanish. This phenomenon aligns with the cost-benefit imbalance characterized by the Kneedle algorithm (Satopaa et al., 2011). Specifically, accuracy for SDv1.4 peaks at 96.7% (5,000 samples) but exhibits a long-tail decline to 95.0% as data increases to 30,000. This validates the 'less is more' principle: minimal data suffice to localize intrinsic forgery traces, whereas excessive samples may lead to overfitting to semantic redundancy. This results in inherent data-efficient robustness, demonstrating the framework's distinct advantage in low-resource settings.

**Robustness to Perturbations.** To evaluate the DNA framework's reliability against real-world image degradations (e.g., social media redistribution), we conducted rigorous stress tests involving Gaussian blurring, JPEG compression, and geometric scaling. As shown in Figure 9, our method maintains exceptional stability across diverse distortions, significantly outperforming methods that rely on fragile surface artifacts. Specifically, even under high-loss compression (JPEG 40), the AP remains robust at 92.4%, while scaling operations show near-perfect invariance with AP consistently exceeding 99%. This evidence shows that the excavated DNA anchors capture robust intrinsic structural anomalies rather than fleeting high-frequency noise, validating the framework's strong potential for practical

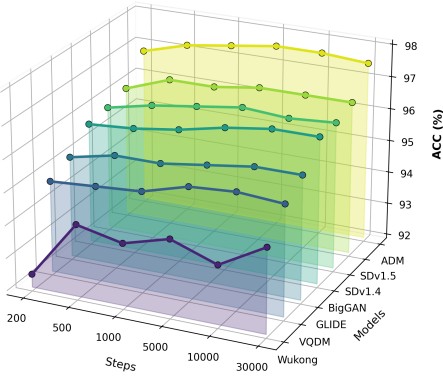

*Figure 8.* **Robustness evaluation under different volumes of training data.** We report the ACC of the DNA framework across seven models.

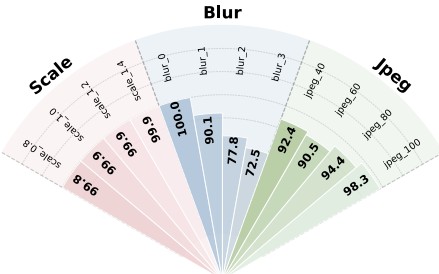

*Figure 9.* **Robustness evaluation under diverse degradation conditions.** We report the AP of the DNA framework across four dimensions: Blurring, JPEG Quality, and Scaling.

deployment in complex scenarios. This comprehensive robustness further validates the potential of our method for deployment in complex, practical scenarios.

**Impact of Pre-training Data Corpora.** To investigate whether FDUs performance stems from the memorization of generative patterns encountered during pre-training (i.e., data leakage), we conducted a chronological isolation experiment using models trained in the "pre-AI era". We evaluated two pristine control groups: (1) Absolute pure, consisting of ResNet-50 (He et al., 2015) and CLIP-B/32 (Radford et al., 2021) trained on the closed-loop ImageNet-1K dataset; and (2) Highly pure, featuring foundation models (CLIP-Base (Radford et al., 2021), DINOv2-Base (Oquab et al., 2024)) trained on web-scale data prior to the proliferation of AIGC. As shown in Figure 10, even without exposure to modern synthetic images, these models achieved exceptional results, with ResNet-50 and DINOv2 reaching 95.4% and 93.6% AP, respectively. This evidence confirms that FDUs do not rely on memorizing specific forgery traces, but rather leverage an intrinsic discriminative instinct developed through general visual representation learning.

**Universality Across Architectures.** To test scalability, we evaluated DNA across diverse foundation models. As shown in Figure 11, DNA consistently improves performance across CNN, ViT, and LLM (Borile & Abrate, 2025)

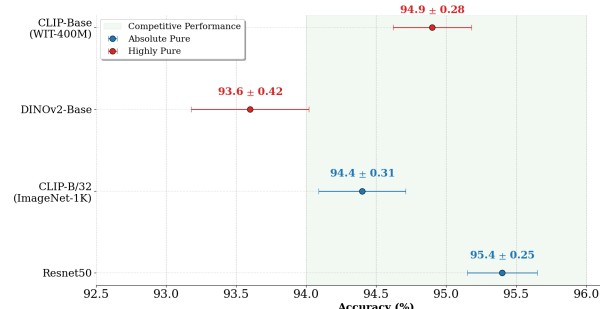

*Figure 10.* **Origin of detection capability.** Models trained on "pure" pre-AIGC datasets (ImageNet-1K and web-scale corpora) high AP, proving that detection is an intrinsic property of general visual understanding rather than a result of generative data leakage.

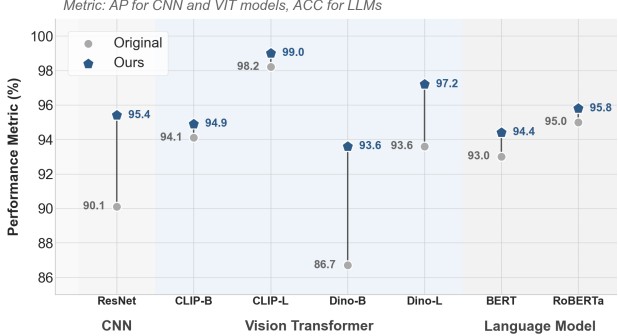

*Figure 11.* **Comparison of performance across various pre-trained backbones.** We assess the generalizability of the proposed method across CNN, ViT, and LLM architectures.

architectures, notably boosting ResNet to 95.4% and Dino-B to 93.6%. This confirms that forgery detection is an emergent property of large-scale pre-training: as models handle more complex tasks, they naturally develop an internal "instinct" for authenticity that DNA successfully extracts.

## 6. Conclusion

This paper challenges the prevailing fine-tuning paradigm by proposing the discriminative neural anchors (DNA) framework, demonstrating that forgery detection is latent knowledge inherent in pre-trained models. By excavating "dormant" sparse neurons within intermediate layers, DNA effectively decouples forgery traces from semantic interference. Extensive evaluations on HIFI-Gen demonstrate DNA's superior few-shot generalization and robustness against cutting-edge models like FLUX and SDv3.5, surpassing full fine-tuning methods. Ultimately, this work not only validates the "less is more" principle but also establishes a new defensive paradigm where "mining intrinsic representations" prevails over "external knowledge injection". We demonstrate that awakening the dormant generalization instinct within models is key to countering evolving generative AI. This finding paves the way for developing more efficient and universal safety mechanisms for foundation models.

## Acknowledgements

This research is supported by the National Research Foundation, Singapore under its National Large Language Models Funding Initiative (AISG Award No: AISG-NMLP-2024-002). Any opinions, findings and conclusions or recommendations expressed in this material are those of the author(s) and do not reflect the views of National Research Foundation, Singapore.

## Impact Statement

This study proposes an innovative neuron-level excavation framework, discriminative neural anchors (DNA), that fundamentally challenges the resource-intensive fine-tuning paradigm by "awakening" the intrinsic forgery-detection knowledge latent within pre-trained foundation models. By precisely isolating sparse forgery-discriminative units (FDUs), this technology achieves superior detection accuracy and generalization even under few-shot conditions. This framework significantly enhances the security of the digital information ecosystem against hyper-realistic AI-generated content while maintaining exceptional computational efficiency. It provides an efficient, interpretable, and universal technical solution for constructing more robust and trustworthy industrial-grade authentication mechanisms.

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

## Supplementary Material

The appendices provide supplementary material and theoretical foundations that support the main paper's findings. Appendix A demonstrates that the identified Forgery-Discriminative Units (FDUs) are essential for detection. Appendix B details the multi-metric intersection strategy for critical layer localization and the curvature-based thresholding used to identify the optimal FDU subspace via the Kneedle algorithm. Appendix C tracks the "discriminative awakening" and "semantic collapse" trajectories across the full 24-layer attention heatmaps. Appendix D presents the comprehensive dataset and technical configurations for the HIFI-Gen benchmark. Appendix E offers an in-depth analysis of robustness to real-world image perturbations. Appendix F provides a comparative efficiency analysis, documenting that the DNA framework achieves an average speedup of more than 10x in inference time compared to baseline methods. Appendix G investigates the impact of pre-training data corpora through chronological isolation experiments, proving that the model's discriminative instinct is independent of AIGC data leakage. Appendix H addresses common issues. Appendix I delivers a formal theoretical justification for the DNA framework, including a proof of the Bayes optimality of sparse mean shifts and a mathematical derivation of the monotonicity of masking impact on classification performance. Appendix J details the limitations and future work.

## A. FDUs Validation

**Monotonic Decline Test.** As shown in Figure 12, as the proportion of removed FDUs increases from 1% to 100%, ACC and AP metrics exhibit a sharp monotonic decline while EER rises steadily. This pronounced monotonicity corroborates that FDUs are the essential components carrying the knowledge critical for forgery detection. Overall, these results indicate that pre-trained models inherently contain a core set of specialized, robust, and universal neurons for authenticity discrimination.

*Figure 12.* **Impact of FDUs masking ratio on detection performance.** Calculating metrics of ACC, AP, and EER as the masking (zero-out) ratio of FDUs increases from 1% to 100%.

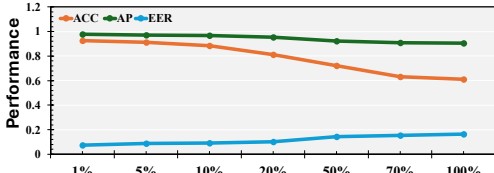

## B. Detailed Methodology & Algorithm

### B.1. Coarse-grained Layer Localization Strategy

In this section, we provide the mathematical formulation and algorithmic details for the coarse-to-fine excavation mechanism. We specifically detail the multi-metric intersection strategy used for layer selection and the curvature-based thresholding algorithm for identifying the optimal FDUs subspace. To systematically identify the "critical layers" ($L_{critical}$) where the model's focus transitions from global semantics to local forgery artifacts, we employ a multi-metric intersection strategy. This strategy integrates geometric separability, attention distribution shifts, and linear classification capability.

Let $\mathcal{L} = \{1, 2, \dots, L\}$ be the set of all transformer layers. We define three subsets of candidate layers based on distinct probing metrics:

**Feature separability candidate** ($L_{sep}$) This set is derived from the cosine distance metric $D_{cos}(i)$ ((2) in the main text). It identifies layers where the representation vectors of real and fake images become orthogonal. A layer $i$ is selected if its separability exceeds the statistical upper bound:

$$L_{sep} = \{i \in \mathcal{L} \mid D_{cos}(i) > \mu_{D_{cos}} + \alpha \cdot \sigma_{D_{cos}}\}, \tag{8}$$

where $\mu_{D_{cos}}$ and $\sigma_{D_{cos}}$ represent the mean and standard deviation of cosine distances across all layers, and $\alpha$ is a scaling factor (empirically set to $1.0$). This ensures we select layers with significantly higher-than-average separability.

**Attention shift candidate** ($L_{attn}$) Derived from the euclidean distance of attention maps $D_{L2}(i)$ ((4) in the main text), this metric highlights layers where the model's spatial attention drastically diverges between real and fake inputs. We identify layers that exhibit local maxima in the attention difference curve:

$$L_{attn} = \{i \in \mathcal{L} \mid D_{L2}(i) > D_{L2}(i-1) \wedge D_{L2}(i) > D_{L2}(i+1)\}. \tag{9}$$

**Linear probing candidate** ($L_{prob}$) This set ensures the selected layers contain sufficient information for a linear classifier to function. Based on the probing accuracy $ACC_{lin}^{(i)}$ (Sec 4.2), we select layers that achieve performance close to the optimal

peak:

$$L_{prob} = \{i \in \mathcal{L} \mid ACC_{lin}^{(i)} \geq \gamma \cdot \max_{j \in \mathcal{L}}(ACC_{lin}^{(j)})\}. \tag{10}$$

The final critical layer interval is determined by the intersection of these functional candidates. This rigorous filtering ensures that selected layers ($L_{critical}$) possess both high linear discriminative power and distinct artifact-sensitive attention mechanisms, while filtering out shallow layers (dominated by noise) and deep layers (dominated by semantic collapse):

$$L_{critical} = L_{sep} \cap L_{attn} \cap L_{prob}. \tag{11}$$

### B.2. Fine-grained FDUs Construction via Kneedle Algorithm

After localizing the critical layers, we compute the tri-factor fusion score $S_{i,k}$ for all neurons. A core challenge is determining the optimal number of neurons to retain without manual threshold tuning. We employ the Kneedle Algorithm (Satopaa et al., 2011) to identify the "elbow point" of the neuron ranking curve, which mathematically represents the point of diminishing returns.

Consider the set of all candidate neurons in layer $l \in L_{critical}$, sorted by their importance scores in descending order. Let $\mathcal{S} = \{s_1, s_2, \ldots, s_N\}$ be the sorted scores, where $N$ is the total number of neurons in that layer. The selection process proceeds in three steps:

**Normalization** Since the absolute values of fusion scores may vary across layers, we map both the rank indices $x$ and the scores $y$ to the unit square $[0,1] \times [0,1]$ to make the curvature analysis scale-invariant:

$$x_k = \frac{k-1}{N-1}, \quad y_k = \frac{s_k - s_{min}}{s_{max} - s_{min}}, \tag{12}$$

where $s_{min}$ and $s_{max}$ are the minimum and maximum scores in $\mathcal{S}$.

**Difference curve calculation** We define a reference chord connecting the start point $(0, y_1)$ and the end point $(1, y_N)$ of the normalized curve. The difference function $D(k)$ calculates the vertical distance from the data point $(x_k, y_k)$ to this chord. This distance effectively measures the "curvature" or the deviation from a uniform distribution:

$$D(k) = y_k - (y_1 + (y_N - y_1) \cdot x_k). \tag{13}$$

**Elbow point identification** The optimal cutoff index $k^*$ (the Elbow Point) is defined as the index that maximizes this difference function. As shown in Figure 13, this point structurally separates the curve into two distinct regions: the "head" (sparse, high-contribution FDUs) and the "long tail" (redundant, semantic neurons).

$$k^* = \operatorname*{argmax}_{k \in \{1, \ldots, N\}} D(k). \tag{14}$$

Finally, the forgery-discriminative units (FDUs) are defined as the top-$k^*$ neurons: $\mathcal{F}_{DNA} = \{n_1, n_2, \ldots, n_{k^*}\}$. This dynamic thresholding ensures that the model adapts the size of the subspace according to the sparsity of forgery traces in each layer.

## C. Qualitative Visualization

This section provides a qualitative analysis of the internal discriminative logic evolution by performing a full 24-layer attention visualization of the forgery-discriminative units (FDUs) identified by the DNA framework, thereby validating the "discriminative awakening" hypothesis. As illustrated in the 24-layer attention heatmaps, the model's discriminative logic follows a typical "U-shaped" trajectory.

A comparison across different layers further reveals that FDUs exhibit remarkable content desensitization. They do not prioritize semantic information, such as "whether this is a fish", but instead focus on artifactual features, such as "whether this part of the pixels is natural". Their activation points pinpoint the logical flaws of generative models in handling complex lighting or geometric structures. This highly concentrated "neural probe" behavior strongly refutes the hypothesis that the model relies on "color bias" or simple "statistical shortcuts" for discrimination. As shown in Figure 14, by qualitatively tracking FDUs attention across 24 layers, we not only visualize the dynamic process of "discriminative awakening" but also demonstrate that the transition from global searching to local precision locking is the core reason our method maintains exceptional robustness in few-shot settings.

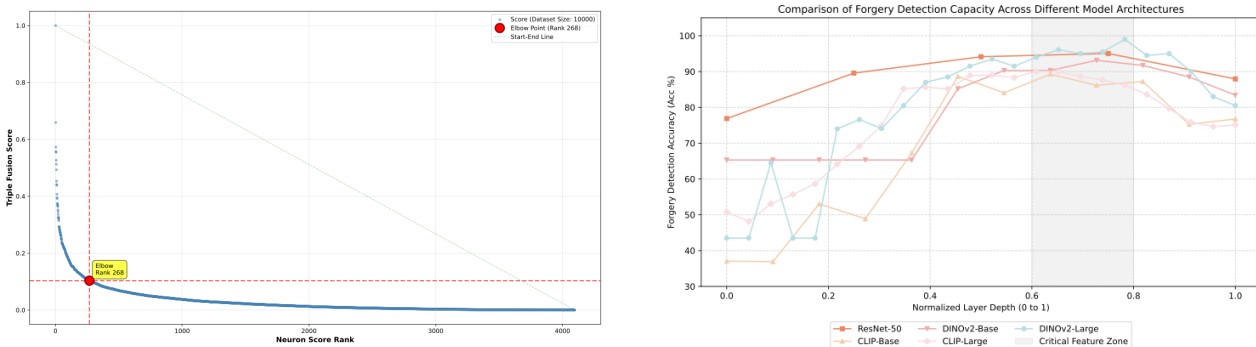

Figure 13. **Analytical results of neuron selection and detection performance.** The left panel illustrates the neuron score distribution and elbow point detection; scores are ranked in descending order. The right panel compares forgery detection accuracy across different model architectures and layer depths.

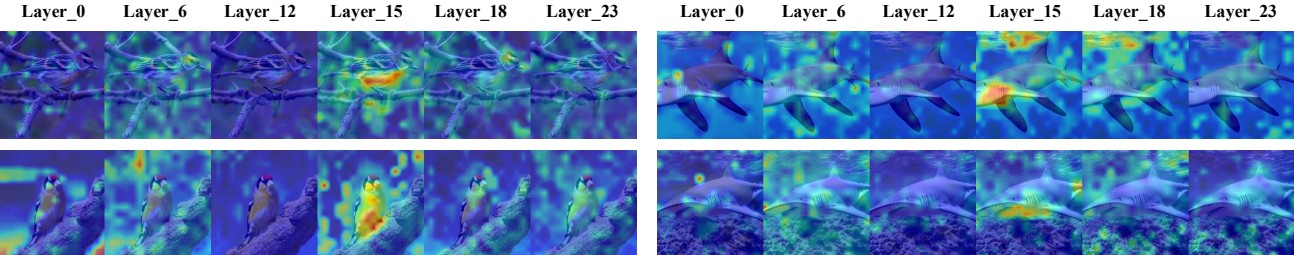

Figure 14. **FDU Attention Across 24 Layers.** This visualization tracks the "discriminative awakening" trajectory of identified Forgery-Discriminative Units (FDUs) across the full depth of the pre-trained backbone.

## D. HIFI-Gen Dataset & Configuration

The construction of the HIFI-Gen dataset aims to extend the existing GenImage benchmark by incorporating state-of-the-art generative models, thereby addressing the forensic research gap regarding emerging generation technologies. This dataset strictly adheres to the GenImage construction, utilizing the ImageNet WordNet ID (wnid) system as its semantic framework. For the generation strategy, we uniformly employed the prompt "photo of class", where the class label is derived from the first entry of the index category. For instance, for the goldfish category, the prompt was fixed as "photo of goldfish". To ensure the benchmark's diversity and challenge, we covered all 1,000 ImageNet categories, with each generator producing 3 test samples per class. Regarding the storage architecture, the dataset is organized strictly by model name within the val directory, which is further partitioned into ai subfolders to store the forged images. In terms of technical parameters, HIFI-Gen integrates five representative, cutting-edge generative architectures, and, crucially, all test samples were produced strictly in accordance with the official meta-settings and recommended best-practice parameters for each model. Specifically, FLUX.1-dev utilized a $1024 \times 1024$ resolution, 28 inference steps, the FlowMatch Euler sampler, and a CFG Scale of 3.5; Stable Diffusion 3.5 was configured with a $1024 \times 1024$ resolution, 40 inference steps, the DPM++ 2M Karras sampler, and a CFG Scale of 5.0; SDXL (Base 1.0) employed a $1024 \times 1024$ resolution, 50 inference steps, the Euler a sampler, and a CFG Scale of 7.5; Stable Diffusion 2.1 used a $768 \times 768$ resolution and 50 inference steps with the DPM++ 2M SDE sampler and a CFG Scale of 7.5; and Z-Image (developed by the Qwen team) was set to a $512 \times 512$ resolution, 30 inference steps, the DDIM sampler, and a CFG Scale of 7.0. Except for SDv2.1, which utilized a sequential offset seed strategy, all other models employed random seeds to enhance image stochasticity. Furthermore, rigorous quality filtering rules were applied to exclude samples with generation failures or significant semantic misalignment. File naming follows the [class_num]_[model]_[index].png format to facilitate efficient retrieval. The HIFI-Gen dataset will be released as an open-source resource to provide a robust benchmark for studying universal image detection under the latest generative paradigms.

# E. Robustness to Perturbations

To further evaluate the utility of the DNA framework within real-world internet ecosystems, such as secondary redistribution on social media and cross-platform re-compression. This section details our stress-testing procedures against various common image degradations. The experimental setup encompasses a broad spectrum of degradation gradients, including Gaussian blurring with kernel sizes $\sigma \in [0, 3.0]$, JPEG compression with Quality Factors (QF) ranging from 100 to 30, and geometric scaling from $0.8\times$ to $1.4\times$. These tests are specifically designed to simulate extreme distortion scenarios encountered during multiple rounds of transmission and processing. Our empirical results demonstrate that the DNA framework not only excels in conventional settings but also maintains exceptional detection precision under intense pressure.

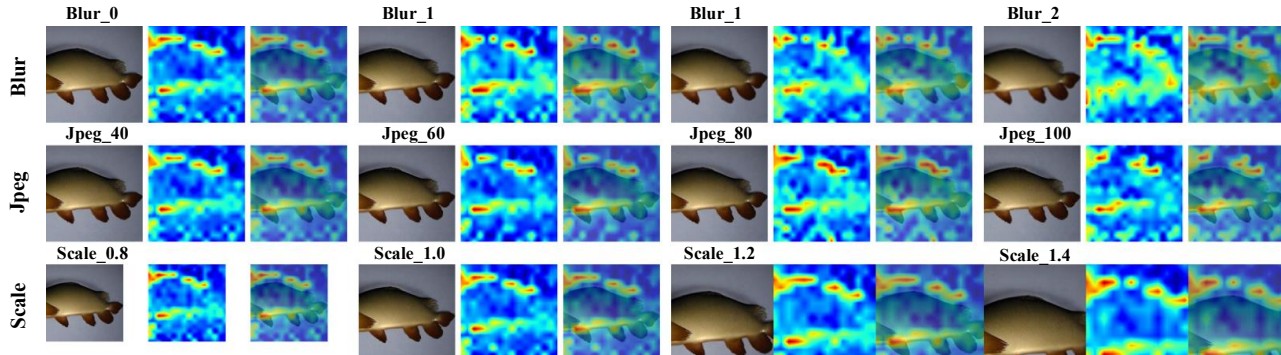

*Figure 15.* **Visualization of attention maps for the DNA framework under perturbations.** We evaluate the stability of attention distribution across different levels of Gaussian blur (Blur), JPEG compression (Jpeg), and resizing (Scale). Each group presents the processed input, the predicted attention map, and their overlay.

The DNA framework exhibits exceptional resilience against diverse stress tests by excavating DNA anchors hidden deep within the pixel layers, thereby capturing intrinsic structural anomalies generated during the synthesis process. These anomalies function as "skeletal" features of the image, possessing a formidable resistance to external interference or attacks. As shown in Figure 15, even in extreme scenarios where visual quality is severely compromised, the heatmaps extracted by our framework maintain high structural consistency. This finding provides compelling evidence that our method does not rely on capturing fleeting, high-frequency noise. Instead, it identifies the inherent and indelible structural fingerprints characteristic of generative models.

# F. Efficiency and Computational Cost Analysis

To evaluate the practical operational efficiency of the DNA framework, we conducted end-to-end inference-time comparison experiments on a unified benchmark dataset of 1,000 images. To ensure a fair comparison, all models were executed under identical hardware configurations. As illustrated in Figure 16, our method significantly reduces computational overhead while maintaining a substantial lead in detection accuracy.

In terms of detection performance, the DNA framework demonstrates superior generalization across a hybrid dataset that includes several mainstream generative models, including ADM, BigGAN, SDv1.4, SDv1.5, and Midjourney. For instance, on the Midjourney test set, the baseline method (MoLD) achieved an accuracy of only 64.1%, whereas our approach reached 97.9%. Regarding Mean Accuracy (Mean Acc), our method outperformed the baseline by a wide margin, scoring 98.6% compared to the baseline's 84%. These results indicate that DNA anchors capture cross-model generative traces more robustly than traditional features.

Crucially, the DNA framework has a significant advantage in processing speed. Using images generated by the ADM model as an example, the baseline method took 13,747.81 seconds to process 1,000 images, while our method took only 595 seconds. Across all tested model categories, our inference time consistently remained within a few hundred seconds, achieving an average speedup of more than $10\times$. This exceptional computational efficiency suggests that the DNA framework can be seamlessly integrated into real-time monitoring systems requiring high throughput, providing a feasible technical solution for the governance of large-scale synthetic imagery on social media.

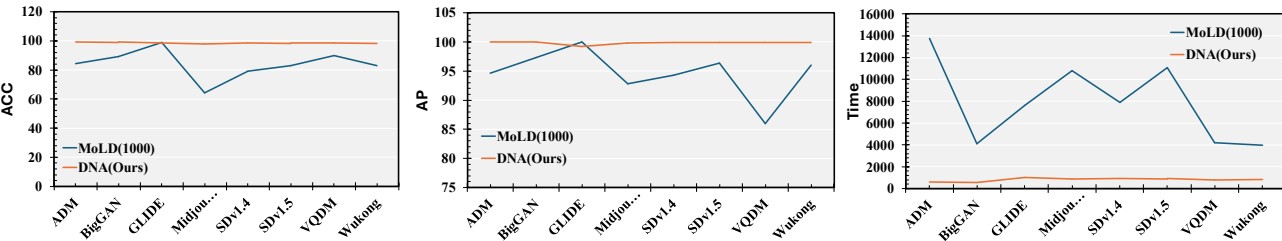

*Figure 16.* **Comparative Analysis of Detection Performance and Computational Efficiency.** This figure illustrates the performance of the DNA framework compared to the MoLD baseline across a dataset of 1,000 images, specifically evaluating Accuracy (ACC), Average Precision (AP), and Time Efficiency.

## G. Impact of Pre-training Data Corpora

To prevent potential data leakage where foundational models might simply "memorize" generative features from their pre-training corpora, we designed a rigorous chronological isolation experiment. We categorized the control groups into **"Absolute Pure" and "Highly Pure"** based on the following logic and comparative value. We divided the control group into two categories: "Absolutely Pure" and "Highly Pure," with the classification logic and comparative value as follows.

The first group is the absolutely pure group, in which the model was trained on closed-loop offline data. Representative models include ResNet-50 and CLIP-B/32 (based on ImageNet-1K training versions). ImageNet-1K is a highly controlled, manually annotated, and closed-loop dataset that was finalized before the AIGC boom (before 2012). Since this dataset contains no images generated by modern diffusion models or large-scale GANs, models trained on it have no chance of encountering "modern forgery traces." If such models can still identify forgeries, it means that the detection capability does not stem from "memorizing" a specific generation algorithm, but rather from an extreme form of modeling the statistical regularities of the real world. The second group is the highly pure group, which was trained using large-scale data from the pre-AIGC era. While these models use large-scale web-crawling data, we selected versions whose data collection and training were completed before the proliferation of generative AI (second half of 2022). This ensures that even if a very small number of early GAN images exist in their pre-training corpora, their proportion is far too insufficient to support model generalization to current Diffusion or DiT architectures. This group aims to test the scale effect of general representation learning on the catalytic effect of "discriminative instinct."

The experimental results demonstrate that even models trained on highly pure datasets maintain high accuracy and discriminative capability. This exceptional performance provides strong evidence for our Discriminative Awakening hypothesis: a deep understanding of the real world naturally encapsulates the ability to identify non-natural creations. This capability is not a skill acquired post-hoc through specific forgery training, but can be effectively unlocked or "awakened".

## H. More Discussions

▷ *Q1. Can DNA generalize to entirely unseen generation architectures, such as the latest DiT or Flow-matching models?*

DNA does not rely on pattern memory; instead, it extracts the model's "instinctive understanding" of statistical patterns in reality. On the HIFI-Gen dataset, which contains FLUX and SDv3.5, DNA maintained an average accuracy of 96.4% without any fine-tuning, demonstrating its ability to uncover universal forgery traces.

▷ *Q2. How do you prove that the identified Forgery-Discriminative Units (FDUs) are essential for the detection task?*

We devised the Monotonic Decline Test. When FDUs were masked proportionally, detection accuracy declined sharply (from 93.1% to 65.1%), whereas randomly masking an equivalent number of neurons had a negligible impact on performance.

▷ *Q3. Is the detection capability a result of generative data leaking into the pre-training datasets?*

We ruled out this possibility through a 'temporal backtracking isolation experiment'. A ResNet-50 trained on fully closed-loop data prior to 2012 still achieved 95.4% AP, demonstrating that detection capabilities stem from a 'redundant instinct' for modelling the real world, rather than a posteriori memory of generated images.

▷ *Q4. What are the specific contributions of gradient sensitivity, activation magnitude, and weight values in your scoring metric?*

These three elements are complementary: activation reflects response strength, weight reflects global contribution, and

gradient captures sensitivity to spurious signals. This tripartite strategy ensures that selected neurons possess both statistical significance and functional specificity.

▷ **Q5. How stable is the "Elbow Point" identification across different backbone architectures?**

The Kneedle algorithm constitutes a parameter-free dynamic adjustment mechanism. As shown in Figure 11, experimental evidence demonstrates its robustness in consistently partitioning neurons into "high-contribution heads" and "redundant tails" across CNNs, ViTs, and even LLMs such as BERT, exhibiting exceptional architectural adaptability.

▷ **Q6. How does DNA perform when facing image degradations common on social media, such as heavy JPEG compression?**

DNA extracts deep-seated 'structural features' rather than fragile surface-level high-frequency noise. Even under JPEG 40 compression, AP maintains 92.4% accuracy and demonstrates exceptional robustness in scaling and blurring tests (Figure 15).

▷ **Q7. Is the "Discriminative Awakening" hypothesis applicable to other domains like AI-generated text detection?**

Demonstrates considerable potential. We validated this approach on language models (BERT/RoBERTa), where performance improvements were equally pronounced. This indicates that mining intrinsic representations constitutes a common defence mechanism across large pre-trained models.

## I. Theoretical Analysis

### I.1. Bayes Optimality of Sparse Mean Shift

We posit that the DNA framework identifies critical intermediate layers by localizing a maximum discrepancy between the latent distributions of real and fake images. Let the feature representations (e.g., [CLS] tokens) of real images ($c = 0$) and fake images ($c = 1$) at a specific layer follow class-conditional distributions. For analytical tractability, we assume these distributions are Gaussian with a shared covariance matrix $\Sigma$:

$$P(f|c = 0) \sim \mathcal{N}(\mu_{real}, \Sigma), \quad P(f|c = 1) \sim \mathcal{N}(\mu_{fake}, \Sigma). \tag{15}$$

In a feature space where the covariance is approximately isotropic (as often observed in pre-trained representations), maximizing the mean shift (centroid distance) is equivalent to minimizing the Bayes classification error.

**Bayes Decision Boundary** For binary classification with equal priors, the optimal decision boundary is defined by the log-likelihood ratio, resulting in a hyperplane:

$$w^T f + b = 0, \tag{16}$$

where the weight vector $w = \Sigma^{-1}(\mu_{fake} - \mu_{real})$.

**Error Rate and Mahalanobis Distance:** The minimum Bayes error rate $P_{error}$ is a strictly decreasing function of the Mahalanobis distance $d$:

$$P_{error} = \Phi\left(-\frac{d}{2}\right), \quad d^2 = (\mu_{fake} - \mu_{real})^T \Sigma^{-1} (\mu_{fake} - \mu_{real}). \tag{17}$$

**DNA's Objective** DNA localizes the "Critical Layers" by maximizing $D_{cos}$ and $D_{L2}$. In high-dimensional pre-trained spaces, features are largely decorrelated, meaning $\Sigma$ approaches the identity matrix $I$. Under this condition, maximizing the geometric mean shift $\|\mu_{fake} - \mu_{real}\|$ directly maximizes $d$ and minimizes $P_{error}$. This justifies why linear probing on DNA-localized layers achieves near-optimal detection performance.

### I.2. Monotonicity of Masking Impact

We define the triadic fusion score for the $k$-th neuron in layer $i$ as $S_{i,k} = |\overline{g}_{i,k} \cdot \overline{a}_{i,k} \cdot w_{i,k}|$. The degradation in detection accuracy is a monotonic function of the cumulative importance scores of the masked neurons.

**Logit Perturbation** Consider the linear probe output $z = \sum_{k=1}^{N} w_k a_k + b$. Masking a set of neurons $\mathcal{F}$ is equivalent to setting their activations $a_k = 0$.

**First-Order Taylor Expansion** The change in classification loss $\mathcal{L}$ can be approximated by the magnitude of the product of the gradient and the activation change:

$$\Delta\mathcal{L} \approx \left| \sum_{k \in \mathcal{F}} \frac{\partial \mathcal{L}}{\partial a_k} \cdot \Delta a_k \right| = \left| \sum_{k \in \mathcal{F}} g_{i,k} \cdot a_{i,k} \right|. \tag{18}$$

**Monotonicity Property** Since $S_{i,k}$ integrates gradient sensitivity, activation magnitude, and the statistical contribution $w_{i,k}$ to the decision boundary, the neurons with the highest $S_{i,k}$ carry the highest density of discriminative information.

**Empirical Verification** As shown in Figure 12, the detection metrics (ACC and AP) exhibit a sharp, monotonic decline as the masking ratio of FDUs increases from 1% to 100%. This confirms that $S_{i,k}$ accurately identifies the irreplaceable functional specificity of these neurons, distinguishing them from random noise or high-magnitude redundant features.

## J. Limitation

While the DNA framework has demonstrated superior efficacy and inference efficiency in identifying forged images by awakening latent neurons in pre-trained models, this research currently focuses primarily on distinguishing AI-generated hyper-realistic images. Our experiments are predominantly based on general-purpose vision backbones (e.g., CLIP, ViT) trained on broad natural image datasets. In future work, it would be valuable to explore the generalization capability of the DNA mechanism in more specialized, high-precision domains, such as medical imaging and other expert scenarios.

