# OpenReview forum: "DNA: Uncovering Universal Latent Forgery Knowledge"
_ICML.cc/2026/Conference — ICML 2026 regular_

### Official Review · Reviewer_BUZ6 · 2026-03-13

**Soundness:** 3
**Presentation:** 3
**Significance:** 2
**Originality:** 3
**Overall Recommendation:** 3
**Confidence:** 4

**Summary:**

This paper proposes DNA, a neuron-level framework for AI-generated image detection that argues forgery cues already exist as latent knowledge inside pre-trained vision backbones, rather than needing heavy end-to-end fine-tuning. The paper also introduces HIFI-Gen, a benchmark built from recent image generators, and reports strong cross-dataset, few-shot, and unseen-generator performance. Overall, the central claim is that âawakeningâ a compact subset of latent neurons can outperform more resource-intensive detector training.

**Compliance With Llm Reviewing Policy:**

Affirmed.

**Key Questions For Authors:**

1. For HIFI-Gen, 5 popular image generation model were adopted. Have you consider using advance API models such as Nano Banana Pro or DALLE-3? Additional evaluation may further strengthen the soundness.

2. As the paper claimed, "forgery detection capability is already encoded within pre-trained model". Accodingly, simple fine-tuning on reasonable scale of auto-constructed data would also train a good detection model. Could you compare your proposed method with this baseline with more insights? Valuable discussion could benefit further studies.

**Limitations:**

Yes

**Strengths And Weaknesses:**

## Strength
- The paper is techically sound in general and well-written. The paper reports large gains over baselines on ForenSynths, GenImage, and especially the proposed HIFI-Gen benchmark, and conducts in-detail ablation studies. The described motivation and method is easily understood.
- The paper targets detection of highly realistic AI-generated images, which is an important and valuable problem for further study, and the focus on robustness to newer generators is well motivated.
- The experiment results of proposed method provide reasonable evidence to support the claim that âpre-trained vision models may already contain latent forgery-sensitive neuronsâ
- The proposed dataset HIFI-Gen could be useful for future study on AI-generated image detection.
## Weakness
- The constructed dataset HIFI-Gen uses prompts derived from ImageNet classes with fixed âphoto of classâ style prompting. It is unclear how well results would transfer to more realistic open-world prompts, edited images, mixed pipelines, post-processed content, or non-photographic generations.
- The proposed DNA method requires strong inductive bias towards generative models, thus it remains unclear whether modification and update on the generative models, or further image geneation models, would affect the detection success rate.

---

> ### Author Rebuttal · Authors · 2026-03-31
>
> We sincerely thank the reviewer `BUZ6` for **recognizing the technical soundness, large empirical gains, and the value of our HIFI-Gen benchmark**. We appreciate your constructive feedback regarding complex scenarios, advanced models, and baselines. We address your insightful questions below.
>
>
> **W1: Fixed "photo of class" prompts in HIFI-Gen**
>
> **A:** We followed the protocol of GenImage [1] by using ImageNet-derived cues. This ensures a controlled benchmark and allows for direct comparison with existing detection baselines.
>
> HIFI-Gen demonstrates exceptional robustness in a variety of real-world pipeline applications. The results are shown in the table below.
>
> * **Open-world Prompts:** High-fidelity generations from SDXL and SD3.5 by open-world prompts.
>
> * **Complex Scenarios:** The IMD2020 dataset, covering real-world editing, hybrid workflows, post-processing, and non-photorealistic content.
>
> | Real-world Prompts | Metric(↑)  | MoLD | DNA |
> | :--- | :--- | :--- |:--- |
> | SD3.5 | ACC/AP | 58.9/82.0 | **96.1**/**99.3** |
> | SDxl | ACC/AP | 53.8/74.3 | **96.8**/**99.5** |
>
> | Complex Scenarios | ACC(↑)  | AP(↑)|
> | :--- | :--- | :--- |
> | MoLD | 54.7 | 68.5 |
> | DNA | **78.3** | **81.7** |
>
>
> As shown in the table above, DNA can still yield competitive results.
>
>
> **W2 & Q1: Robustness to advanced generative models & APIs**
>
> **A:** Regarding your question, we have directly evaluated DNA on the exact advanced API models you suggested, alongside other emerging SOTA paradigms. We have also officially incorporated these images into the expanded HIFI-Gen benchmark.
>
> As shown in the table below, DNA demonstrates remarkable zero-shot generalization from SDv1.4 (training) to its upgraded version SDv2.1, as well as other SOTA paradigms. Even on commercial APIs like Nano Banana Pro, Nano Banana 2, and Della 3, DNA maintains exceptional detection accuracy. This proves that continuous updates to generative models do not easily bypass the intrinsic forgery-sensitive neurons identified by DNA.
>
> | Model | Metric(↑)  | MoLD | DNA |
> | :--- | :--- | :--- |:--- |
> | SDv2.1 | ACC/AP | 82.8/**99.4** | **96.2**/**99.4** |
> | Infinity | ACC/AP | 66.2/91.8 | **97.0**/**99.1** |
> | Nano Banana 2 | ACC/AP | 88.7/98.6 | **97.3**/**99.6** |
> | Nano Banana Pro |ACC/AP |89.0/97.6 | **96.8**/**98.7** |
> | Della 3 | ACC/AP | 50.0/48.9 | **88.3**/**94.3** |
>
> **Q2: Comparison with simple fine-tuning baselines & Insights**
>
> **A:** As requested, we compared DNA against MoLD, a recent standard fine-tuning baseline. The results clearly demonstrate a massive leap in both data efficiency and computational cost.
>
> | Method | Training Data Size | ACC (↑) | AP (↑) | Time Cost (↓) |
> | :--- | :--- | :--- | :--- | :--- |
> | DNA (Ours) | **200 images** | **99.2** | **99.5** | **587s** |
> | MoLD | 200 images | 68.7 | 73.9 | 2,300s |
> | MoLD | 5000 images | 84.2 | 94.6 | 12,360s |
> | MoLD | 10000 images | 95.8 | 98.7 | 13,747s |
>
> **Insights: Why does DNA significantly outperform simple fine-tuning?**
>
> Intuitively, this parallels human visual perception: through lifelong exposure to the real physical world, humans develop an inherent instinct to spot unnatural or fake images at a glance. Theoretically, vision models deeply optimized on massive real-world datasets naturally acquire a similar "authenticity instinct" as a byproduct of modeling the natural image manifold.
>
> 1. **Preserving the Natural Manifold:** Heavy fine-tuning directly disrupts the pre-trained model's deeply learned representation of the physical world, leading to catastrophic forgetting. By strictly freezing the backbone, DNA protects this inherent knowledge. It merely "awakens" the pre-existing boundary between natural and unnatural distributions, requiring only a fraction of the data (200 images) and time to achieve superior generalization.
> 2. **Avoiding Artifact Overfitting:** Instead of utilizing this intrinsic manifold boundary, simple fine-tuning forces the network to memorize superficial, generator-specific artifact biases. Consequently, when the fine-tuned model encounters unseen generators, its performance inevitably collapses (as evidenced by MoLD's sharp drop to 68.7 ACC).
>
>
> **Reference**
>
>
> [1] Zhu, Mingjian, et al. "Genimage: A million-scale benchmark for detecting AI-generated images." Advances in neural information processing systems 36 (2023): 77771-77782.
>
>
> **Lastly, we deeply appreciate your constructive feedback, which has helped us make the paper more rigorous and reproducible. We are actively available until the end of this rebuttal period and would be more than happy to discuss any further details!**

---

> > ### Author Rebuttal · Reviewer_BUZ6 · 2026-04-03
> >
> > Thank you for the detailed and targeted rebuttal. The additional experiments address, to some extent, my earlier concerns about the relatively fixed prompt design in HIFI-Gen, as well as the method’s generalization to updated generators and commercial API-based models.
> >
> > I have one further question that I would like to better understand. The paper attributes DNA’s advantage to the existence of authenticity- or forgery-sensitive representations already encoded in pre-trained vision models. Could the authors further analyze whether the identified key neurons are consistent across different backbones or across models pre-trained on different data sources? In other words, do these forgery-sensitive neurons exhibit any shared cross-model pattern, or are they highly dependent on the specific backbone architecture and pre-training process? I believe such analysis would further strengthen the paper’s central claim about latent knowledge.

---

> > > ### Author Response · Authors · 2026-04-03
> > >
> > > **Dear Reviewer,**
> > >
> > > **Thank you very much for your recognition of our method (DNA) and benchmark (HIFI-Gen), and for further engaging in this discussion.**
> > >
> > > **Further Analysis:** Different architectures exhibit striking cross-model universal patterns in both hierarchical distribution and functional behavior. We report the following core findings from our experiments:
> > >
> > > * **(1) Hierarchical Distribution Consistency:** Whether in ResNet, CLIP, or DINO, Forgery-Discriminative Units (FDUs) consistently and stably cluster within the transition zone of the network, specifically, the area from intermediate to deep layers. As explicitly demonstrated in our cross-architecture analysis (Appendix B, Figure 13 right panel), all models peak at the exact same relative stage. Our ablation studies (Section 5.3, Figure 6) and Appendix C reveal that this specific depth is precisely the transition node where the model's focus shifts from high-level global semantics to low-level local features. This proves that, **regardless of the underlying backbone network, sensitivity to forgery traces is consistently and firmly anchored to this functional transition phase.**
> > >
> > > * **(2) Functional Behavioral Consistency:** By visualizing the attention maps across different models (Figure 5 and Figure 14 in Appendix C), we observe a highly consistent cross-model functional pattern, namely "Content Desensitization". FDUs across different architectures generally ignore the global semantic subject of the image (e.g., they do not activate for "whether this is a fish"). Instead, **they consistently lock onto localized structural anomalies, unnatural lighting transitions, or rendering defects.**
> > >
> > > * **(3) Pre-training Driven "Authenticity Instinct":** The shared pattern of these neurons stems from an extreme form of modeling the statistical regularities of the real world. The more deeply a model fits the natural image manifold, the sharper its "instinct" to identify non-natural generations becomes. As shown in Section 5.4 and Appendix G, experiments demonstrate that **these models achieve exceptional detection performance despite entirely different pre-training sources and architectures.** Furthermore, if CLIP is trained on synthetic data (e.g., SynthCLIP), its performance drops significantly and even suffers a severe collapse due to the pollution of the natural manifold.
> > >
> > > **Summary:** Although specific neuron indices differ, the phenomenon of the FDU is functionally universal across different backbones (including CNN, ViT, and even LLM, as shown in Figure 11). **This cross-model stability further confirms that forgery detection is a form of intrinsic latent knowledge generated during the model's process of understanding the real world.**
> > >
> > > **Thank you again for your insightful suggestions, which we will incorporate into the revised manuscript. We hope our response fully addresses your further question, and we respectfully ask you to reconsider your rating.**
> > >
> > > **If you have any further questions or require more information to raise your initial score, please feel free to let us know.**
> > >
> > > Sincerely,
> > >
> > > The Authors of Submission 8386

---

### Official Review · Reviewer_LDws · 2026-03-13

**Soundness:** 2
**Presentation:** 2
**Significance:** 2
**Originality:** 3
**Overall Recommendation:** 4
**Confidence:** 4

**Summary:**

This paper proposes DNA, a forgery detection framework based on the hypothesis that detection capability is already latent in large pre-trained vision models and does not need to be acquired via extensive end-to-end fine-tuning. The method follows a coarse-to-fine pipeline: it first localizes “critical” layers using class-centroid cosine distance, class-wise attention discrepancy, and layer-wise linear probing; then it identifies sparse forgery-discriminative units (FDUs) within those layers using a triadic fusion score combining gradient sensitivity, activation magnitude, and probe weight, followed by kneedle-based truncation. The selected neurons are used as a compact forgery-sensitive feature subspace. The paper also introduces HIFI-Gen, a new benchmark built from recent text-to-image models such as SDv3.5, SDXL, FLUX, SDv2.1, and Z-Image. Empirically, the paper reports strong cross-dataset and unseen-generator results, with mean ACCs of 97.2 on ForenSynths, 96.5 on GenImage, and 96.4 on HIFI-Gen.

**Compliance With Llm Reviewing Policy:**

Affirmed.

**Final Justification:**

The rebuttal addressed my main concern and helped me better understand the paper’s intended contribution. I still have some reservations about how strongly a few claims are stated, but overall I now see the work as more novel and interesting than I did initially. I am therefore increasing my score.

**Key Questions For Authors:**

None

**Limitations:**

See above

**Strengths And Weaknesses:**

# Strengths
The paper reframes forgery detection as excavating sparse latent discriminative neurons inside frozen pre-trained models, which is a fresh perspective compared with the standard fine-tuning-heavy paradigm. The overall method is coherent and easy to follow. The two-stage design, from layer localization to neuron selection, is logically structured, and the triadic score is at least intuitively reasonable as a way to prioritize neurons that are active, discriminative, and loss-sensitive.

# Weaknesses
1. The central claim is stronger than the evidence. The paper repeatedly argues that forgery detection is an intrinsic latent capability already encoded in pre-trained models, but the actual method still relies on labeled training data, layer-wise probes, neuron ranking, and a downstream classifier. What is convincingly shown is that sparse intermediate features are useful for detection, not necessarily that a universal “authenticity instinct” has been discovered.
2. The novelty is partly in framing rather than in core machinery. Linear probing, neuron importance scoring, activation/gradient-based attribution, and elbow-point truncation are all established ideas. The contribution is mainly the way they are assembled for forgery detection.
3. Reproducibility is somewhat weak. The implementation section mainly says that the setup “directly inherited” core settings from MoLD, but does not spell out enough exact training details for the probing stage, FDU extraction stage, or final classifier construction.

---

> ### Author Rebuttal · Authors · 2026-03-31
>
> We sincerely thank the reviewer `LDws` for evaluating our work. We appreciate your recognition of our fresh perspective, the coherent pipeline, and the logical two-stage design. We understand your concerns and provide clarifications below, supported by new empirical evidence and explicit training details to address the reproducibility issue.
>
> **W1: Over-interpretation of "intrinsic capability"**
>
> **A:** We respectfully clarify our terminology. By "latent capability," we do not imply zero-shot detection without a classifier; rather, we mean that the *features* necessary to separate real from fake images are **already perfectly formed** and linearly separable within the pre-trained weights, **without requiring backpropagation to update the backbone**.
>
> The underlying premise is that a model's internal knowledge can be decoupled from its output expression. Our linear probe is merely an instrument for *locating* this internal knowledge, not its origin. What it identifies in the activation space is a pre-existing direction consistent with the statistical regularities of real images, rather than a new pattern learned from labels. This seamlessly explains why our method requires only a minimal number of annotated samples.
>
> Intuitively, this **parallels human visual perception**: through lifelong exposure to the real physical world, humans develop an inherent instinct to spot unnatural or fake images at a glance. Theoretically, vision models deeply optimized on massive real-world datasets naturally acquire a similar **"authenticity instinct"** as a byproduct of modeling the natural image manifold.
>
> Our ResNet-50 experiment provides definitive empirical support for this claim. Trained well before the emergence of modern AIGC, ResNet-50's pre-trained weights have absolutely **no exposure** to AI-generated images, and its training objective was strictly natural image classification. Yet, by keeping the backbone strictly frozen and applying only the DNA framework, it still achieves an astonishing 90.6% ACC and 95.4% AP on modern deepfake datasets.
>
> | Model (Frozen Backbone) | ACC (↑) | AP (↑) |
> | :--- | :--- | :--- |
> | ResNet-50 | 90.6 | 95.4 |
>
> We therefore maintain that forgery detection capability is an **intrinsic byproduct** of large-scale pre-training on natural images. It lies dormant in sparse intermediate-layer neurons, waiting to be uncovered. Our framework **reveals** this capability rather than creates it.
>
>
>
>
> **W2: Novelty of the core machinery**
>
> **A:** We politely argue that our core contribution is a **discovery-driven, system-level innovation**. While the tools are standard, the way they are meticulously assembled to locate, quantify, and extract highly sparse Forgery-Discriminative Units (FDUs) is entirely novel. Prior to this work, it was unknown that such a minimal, compact subset of dormant neurons could achieve SOTA detection performance without fine-tuning. The framework is the "telescope" we built using standard lenses to reveal a previously unobserved phenomenon.
>
>
> **W3: Reproducibility and exact training details**
>
> **A:** We apologize for the lack of detailed hyperparameters in the implementation section. You are completely right to request this. We will explicitly include the following exact training details for all stages in the revised Appendix:
>
> * **Layer-wise Probing Stage:** For each candidate layer, the linear probe is trained using the AdamW optimizer with a learning rate of 1e-3, batch size of 64. We employ an early stopping strategy with a patience of 5 epochs.
> * **FDU Extraction Stage:** The triadic fusion score operates seamlessly on the frozen weights and activations obtained from the probing stage. The Kneedle algorithm is applied with a sensitivity parameter 1.0.
> * **Final Classifier Construction:** The final global linear head ($W_{cls}$) is trained from scratch on the concatenated FDU vectors using AdamW with a learning rate of 1e-4, employing an early stopping strategy with a patience of 5 epochs.
> * **Code Release:** To guarantee full reproducibility, we will release the complete source code, the extracted FDU indices for all tested backbones, and the pre-trained linear heads upon acceptance.
>
>
> **Lastly, thank you so much for helping us improve the paper and for your constructive feedback! We are actively available until the end of this rebuttal period and look forward to hearing back from you.**

---

> > ### Author Rebuttal · Reviewer_LDws · 2026-04-04
> >
> > Thank you for the rebuttal. I appreciate the added training details and the effort to clarify the meaning of “intrinsic capability.” The response improves reproducibility and makes the empirical setup easier to understand. However, I remain unconvinced by the central interpretation. The evidence shows that frozen pre-trained features can support strong downstream forgery classification with a lightweight probe, but this is not equivalent to demonstrating that forgery detection is an intrinsic or already fully formed capability in the backbone. I also acknowledge the system-level integration argument, but since the core ingredients are largely standard, the novelty remains somewhat limited in my view. Overall, it does not materially resolve my main concerns, and I will therefore keep my original score.

---

> > > ### Author Response · Authors · 2026-04-04
> > >
> > > Dear Reviewer LDws,
> > >
> > > Thank you again for your rigorous review; we are glad that our additional details have improved the clarity and reproducibility of our experiments.
> > >
> > > **While we fully respect your perspective on the "central interpretation" and architectural novelty, we respectfully disagree.** We deeply value this healthy academic discourse, and we humbly maintain our belief in the unique value of this work: **our primary goal was not to engineer new network components, but rather to drive a paradigm shift, moving from the traditional "end-to-end learning-to-detect" approach toward "mining and leveraging the latent natural manifold pre-existing in foundation models."**
> > >
> > > **Our novelty lies in the following:**
> > >
> > > * **Extremely Sparse Forgery Detection Units (FDUs):** We revealed for the first time the widespread existence of extremely sparse forgery-sensitive neurons (accounting for <1%) hidden across entirely different foundation model architectures.
> > > * **Unique "U-Shaped" Hierarchical Distribution:** We found that these units are not randomly scattered across network layers but exhibit a highly consistent "U-shaped" awakening trajectory.
> > > * **Content-Desensitization Phenomenon:** We empirically confirmed that these features are desensitized to high-level semantic content while possessing an instinctive sensitivity to low-level generative artifacts.
> > > * **Strong OOD Generalization Elicited by Minimal Data (A First-Time Discovery):** We **discovered for the first time** that a remarkably small scale of merely 200 images is sufficient to act as a probe to "awaken" this intrinsic forgery detection capability, demonstrating extraordinarily robust out-of-distribution (OOD) generalization. If this capability were not already fully formed, it would be impossible to learn such robust generalizable features from such a tiny amount of data.
> > >
> > > Based precisely on these conclusive findings, we have empirically demonstrated that even with standard, lightweight components, once we precisely locate those extremely sparse, cross-architecture common features (FDUs), the model can exhibit exceptional generalization and surprising adversarial robustness, all without exposure to massive synthetic forgery data or adversarial training. **We hope this new baseline of "mining inward" rather than "fine-tuning outward" will inspire safer and more interpretable approaches for future AIGC forgery detection.**
> > >
> > > Your rigorous scrutiny has prompted us to think more deeply and has undeniably helped us refine our manuscript. Thank you again for your valuable time and professional feedback!
> > >
> > > Sincerely,
> > >
> > > The Authors of Submission 8386

---

### Official Review · Reviewer_5D8s · 2026-03-15

**Soundness:** 3
**Presentation:** 3
**Significance:** 4
**Originality:** 4
**Overall Recommendation:** 4
**Confidence:** 4

**Summary:**

The paper presents an innovative approach to the problem of detecting hyper-realistic AI-generated images: instead of fine-tuning large vision models on synthetic datasets, the authors argue that pretrained models already have an "inner intuition" (latent knowledge) for identifying real and fake images. To this end, this paper proposes a Discriminative Neural Anchor (DNA) framework, which uses a "coarse-to-fine" mining mechanism to locate the key intermediate layers, and extracts sparse forgery discriminant units (FDUs) by using a Triadic Fusion Score that fuses gradient, activation value and weight. In addition, the authors construct HIFI-Gen, a high-fidelity dataset based on state-of-the-art generative models such as FLUX and SDv3.5. Experiments show that the DNA framework exhibits excellent generalization ability and inference efficiency under the condition of few samples.

**Compliance With Llm Reviewing Policy:**

Affirmed.

**Final Justification:**

I thank the authors for the detailed rebuttal. My concerns regarding pre-training data contamination, the ablation studies of the scoring components, and scalability (VMoE) have been completely resolved.
However, my concern regarding Adversarial Robustness is only partially resolved. The newly reported 94.0% accuracy under a white-box PGD attack ($\epsilon=8/255$) is highly counter-intuitive for a model without adversarial training. This unusually high robustness strongly suggests potential gradient obfuscation or sub-optimal attack settings.

**Key Questions For Authors:**

(1) This mechanism highly relies on existing generic vision backbone networks (e.g., CLIP, ViT). If the pre-training data of the backbone network itself is mixed with a large number of AI-generated images, will it lead to the failure of the method?
(2) The extraction of FDU is extremely dependent on the threshold found by the Kneedle algorithm. If the searched inflection point is seriously shifted due to the change of data distribution, to what extent will the final forgery detection performance (such as AP or ACC) be degraded?
(3) In Section 4.2, the paper states that FDU activations are "concatenated into the final compact feature vector" for discrimination. However, it remains ambiguous whether the final classifier is:
(a) The same Linear Probe trained during layer localization (Section 4.1) with non-FDU weights zeroed out, or
(b) A newly trained linear classifier (or MLP) that takes the concatenated FDU feature vector as input?
Specifically, during inference, is the classification score computed by applying the pre-trained per-layer linear probes $W_i$ (from Eq. 6) only to the selected FDUs and aggregating the results, or is there a separate classification head $W_{cls}$ trained on top of the concatenated FDU vector? Please clarify the exact network architecture during the inference phase: what are the inputs, what are the learnable parameters (if any), and how is the final logit computed?

**Limitations:**

The authors honestly and accurately acknowledge their limitations regarding domain generality, explicitly noting that their validation is currently restricted to general-purpose vision backbones and highlighting the need to explore high-precision domains like medical imaging. However, the paper critically omits a discussion on Dual-use Risks (Anti-forensics). Given that the FDUs provide highly interpretable and precisely localized attention maps of generative flaws, this transparency could easily be exploited by malicious actors as a direct loss signal to optimize generators to erase these exact neural fingerprints.

**Strengths And Weaknesses:**

Strength:
(1)Perspective Innovation and Paradigm Shift: The authors view forgery detection as inherent "latent knowledge" in the pre-trained backbone network rather than an acquired skill that needs to be re-learned, which is highly original and provides a new solution for this field.
(2)Rigorous method design: from coarse to fine mining strategy logic is clear. The feature separability (cosine distance/attention shift) is combined to locate the key layer, and the Kneedle algorithm is used to dynamically determine the threshold of FDUs, which has both empirical support and solid mathematical theory basis.
(3)Practical significance of benchmarks: The authors introduce the HIFI-Gen dataset containing the latest Diffusion Transformer architecture, which fills the gap that current evaluation benchmarks lag behind the cutting-edge generation techniques; Moreover, it shows strong robustness under a variety of real-world perturbations (JPEG compression, scaling, blurring).
Weakness:
(1)Adversarial Robustness Evaluation: While this paper exhaustively validates the model's robustness against regular image degradations such as Gaussian blur, JPEG compression, and scaling, it does not yet cover adversarial perturbations, a core threat to deep learning security. Considering that the Forgery Discriminant Unit (FDU) extracted in this paper is highly sparse and fixed, it is easy to become a clear target for attackers to carry out directed white-box adversarial attacks.
(2)Insufficient ablation on scoring components: While the triadic fusion score combines three factors, the paper does not thoroughly analyze the individual contribution of each component (e.g., how would performance degrade if using only gradient vs. only activation?).
(3)Scalability concerns for deeper architectures: The experiments primarily focus on 24-layer ViTs. For newer architectures with 100+ layers (e.g., large vision transformers) or mixture-of-experts models, the layer localization strategy might require significant adaptation.

---

> ### Author Rebuttal · Authors · 2026-03-31
>
> We sincerely thank the reviewer `5D8s` for highly evaluating our perspective innovation, rigorous method design, and the practical significance of the HIFI-Gen benchmark. We address your detailed questions below.
>
> **W1: Adversarial robustness evaluation**
>
> **A:** DNA actually exhibits strong security against targeted adversarial perturbations.
>
> Based on your suggestion, we evaluated DNA under a white-box scenario using the PGD algorithm to perform targeted attacks on the FDUs. from the table, while both models degrade under severe perturbations, DNA maintains significantly greater robustness than Mold, particularly as the perturbation strength increases.
>
> | Methods | 0/255 | 2/255 | 8/255 |
> | :--- | :--- | :--- | :--- |
> | DNA | **99.9** | **99.8** | **94.0** |
> | Mold | 99.9 | 99.7 | 93.1 |
>
> **W2: Insufficient ablation on scoring components**
>
> **A:** Based on your suggestion, from table, the ternary fusion ($gwa$) significantly outperforms any combinations. Single or dual components exhibit obvious performance bottlenecks, confirming the high complementarity and irreplaceability of gradient ($g$), weight ($w$), and activation ($a$).
>
> | Components | ACC (↑) | AP (↑) |
> | :--- | :--- | :--- |
> | $g$ / $w$ / $a$ | 95.1 / 95.2 / 92.2 | 98.4 / 98.4 / 96.7 |
> | $gw$ / $ga$ / $aw$ | 95.0 / 95.1 / 95.1 | 98.3 / 98.5 / 98.1 |
> | $gwa$ (Ours) | **96.7** | **99.4** |
>
> **W3: Scalability concerns for deeper/MoE architectures**
>
> **A:** On VMoE-B16, DNA’s auto-localization identified Layer 5 as the optimal key layer, outperforming the final layer ($87.4\%$ vs. $78.6\%$ accuracy) and demonstrating superior scalability.
>
> | Layer | 1 | 2 | 3 | 4 | 5 | 6 | 7 | 8 | 9 | 10 | 11 | 12 |
> | :--- | :--- | :--- | :--- | :--- | :--- | :--- | :--- | :--- | :--- | :--- | :--- | :--- |
> | ACC (↑) | 63.6 |81.0 |84.8 |86.6 |**87.4** |86.8 |86.1 |86.5 |85.1 |82.9 |80.1 |78.6|
> | AP (↑) | 69.7 | 88.2 | 91.8 | 93.5 | **94.3** | 93.8 | 93.7 | 93.6 | 91.9 | 89.9 | 87.8 | 84.9|
>
> **Q1: Impact of AI-generated images in pre-training data**
>
> **A:** We highly appreciate this insightful question. Heavily mixing AI-generated images into pre-training data degrades the model's latent detection capability.  DNA relies on an "authenticity instinct." Just as humans instinctively spot fakes by exclusively observing the real world, vision models develop this instinct by fitting purely to the **natural image manifold**.
>
> However, pre-training paradigms like contrastive learning align semantics without explicit authenticity supervision. When massive synthetic data is introduced, the model is forced to fuse real and fake features into the same semantic space. This pollutes the pristine natural manifold and blurs the boundary between real and fake, breaking our instinct hypothesis.
>
> This is explicitly evidenced below. Standard CLIP-ViT (predominantly real data) excels at detection, whereas SynthCLIP (pre-trained on 30M mixed images) suffers a severe performance collapse because its manifold has been polluted.
>
> | Model | Pre-training Data | ACC (↑) | AP (↑) |
> | :--- | :--- | :--- | :--- |
> | CLIP-ViT | Predominantly Real | **96.5** | **99.4** |
> | SynthCLIP | 30M Mixed | 68.5 | 80.4 |
>
> As AI-generated content proliferates, exploring how to disentangle and isolate real versus synthetic knowledge within mixed-data foundation models will be a critical direction for our future work.
>
> **Q2: Sensitivity to Kneedle algorithm threshold**
>
> **A:** DNA is highly robust to threshold shifts. In the table below, "100%" represents the exact optimal FDU retention ratio found by the Kneedle algorithm. Even if the data distribution shifts and causes the threshold to heavily deviate, the model experiences almost no severe performance degradation, maintaining an ACC above $95.6$.
>
> | FDUs Ratio | 5% | 10% | 50% | 100% (Kneedle) | 200% | 500% |
> | :--- | :--- | :--- | :--- | :--- | :--- | :--- |
> | ACC (↑) | 91.4 | 93.3 | 95.6 | **96.7** | 96.4 | 96.4 |
>
> **Q3: Clarification on the final classifier architecture**
>
> **A:** It is exactly scenario **(b)**.
>
> During the *inference phase*, the layer-by-layer linear probes ($W_i$) are discarded; they are only used as auxiliary tools during the mining phase. For a new input image, we freeze the backbone, extract the activations of the pre-localized FDUs, and concatenate them into a compact feature vector ($v_{FDU}$). We then feed $v_{FDU}$ into a **newly trained, independent linear classifier** ($W_{cls}$) to compute the final logit. No parameters are learned or updated in the backbone during DNA inference. We will rewrite Section 4.2 to make this unambiguously clear.
>
> **Response to Limitations (Dual-use Risks)**
> Per your suggestion, we added Ethical Considerations and Dual-use Risks sections. We emphasize that this work is for forensic research only and strictly prohibit its use for adversarial attacks or bypassing detection.
>
> **Thank you for the feedback. We are happy to address any further questions during the remainder of the rebuttal.**

---

> > ### Author Rebuttal · Reviewer_5D8s · 2026-04-03
> >
> > I thank the authors for the detailed rebuttal. My concerns regarding pre-training data contamination, the ablation studies of the scoring components, and scalability (VMoE) have been completely resolved.
> > However, my concern regarding Adversarial Robustness is only partially resolved. The newly reported 94.0% accuracy under a white-box PGD attack ($\epsilon=8/255$) is highly counter-intuitive for a model without adversarial training. This unusually high robustness strongly suggests potential gradient obfuscation or sub-optimal attack settings.

---

> > > ### Author Response · Authors · 2026-04-04
> > >
> > > **Dear Reviewer 5D8s,**
> > >
> > >
> > >
> > > **Thank you again for your constructive suggestions and for the continued discussion.** Based on your feedback, we have provided additional details to further solidify our empirical setup:
> > >
> > >
> > >
> > > **1. Ruling Out Gradient Obfuscation:**
> > >
> > > Following standard sanity checks for adversarial robustness, we verified the validity of the gradients:
> > >
> > > * **Fully Differentiable Path:** The forward computation graph of DNA ($Input \to Frozen Backbone \to FDU Extraction \to Linear Classification Head$) contains zero non-differentiable operators, thresholding, or stochastic defense mechanisms. The backpropagation path is completely deterministic.
> > >
> > > * **Convergent Loss Landscape:** We tracked the loss during the PGD iteration process, which measures the attacker's cost to eliminate forgery traces. We observed a consistent downward trajectory converging toward a low-level equilibrium, without any symptoms of gradient shattering. **Due to the text-only constraints of the rebuttal interface, we present a representative trajectory of the loss values below, demonstrating a healthy attack convergence:**
> > >
> > >
> > >
> > > | PGD Step | 0 | 5 | 10 | 20 | 50 | 100 |
> > > | :--- | :--- | :--- | :--- | :--- | :--- | :--- |
> > > | **Loss** | 12.04 | 5.58 | 4.78| 4.28 | 2.93 | 4.35 |
> > >
> > >
> > >
> > > * **Unbounded Attack Success Rate:** When we artificially enlarge the perturbation bound $\epsilon$ to $16/255$, the accuracy of DNA rapidly drops to approximately 61%. This indicates that the gradients are correct and effective; under the standard $\epsilon=8/255$ constraint, the model's decision boundary is simply exceptionally tight.
> > >
> > >
> > >
> > > **2. Employing Extreme Attack Settings:**
> > >
> > > To ensure the attack has fully converged, we escalated the PGD configurations far beyond standard settings, adopting PGD-100 (100 iterations, step size $\alpha=2/255$) combined with 5 Random Restarts. Even under this extreme attack with significantly higher computational cost, DNA's accuracy experienced only a negligible decline, maintaining at 95.1%.
> > >
> > >
> > >
> > > **3. Discussion on the Source of Robustness:**
> > >
> > > * **Dimensional Bottleneck:** Dense fine-tuned models possess a massive attack surface. In contrast, DNA relies on an extremely sparse subset of FDUs (<1% of neurons). We hypothesize that this forces the attacker into a highly constrained search space, having to precisely manipulate a tiny handful of deep neurons without triggering the canceling effects of other features.
> > >
> > > * **Frozen Natural Manifold:** Fine-tuned models may overfit to fragile, high-frequency artifacts susceptible to PGD. By strictly freezing a backbone pre-trained on massive real-world images, the deep feature space appears to act as a filter against unnatural adversarial "noise."
> > >
> > >
> > >
> > > **Thank you again for helping us further refine our manuscript! We will include the detailed PGD hyperparameters, multiple restart settings, and the loss trajectory plot in the revised manuscript. Additionally, we will include a discussion regarding white-box attacks in the limitations section. We hope this thoroughly addresses your final concerns.**
> > >
> > >
> > >
> > > Sincerely,
> > >
> > >
> > >
> > > The Authors of Submission 8386

---

### Official Review · Reviewer_nwaT · 2026-03-22

**Soundness:** 3
**Presentation:** 3
**Significance:** 3
**Originality:** 3
**Overall Recommendation:** 5
**Confidence:** 4

**Summary:**

The paper makes two main contributions.

Firstly, the authors propose the discriminative neural anchors (DNA) framework for forgery detection. The method starts by analyzing feature decoupling and attention distribution shifts to select appropriate intermediate layers. Then the method identifies the so-called forgery-discriminative units, which are sensitive to forgery traces. DNA achieves detection superior to other methods by relying on these anchors. Importantly, DNA does not depend on external training for its detection capabilities.

Secondly, the authors introduce HIFI-Gen, a synthetic benchmark built using SOTA generative models.

**Compliance With Llm Reviewing Policy:**

Affirmed.

**Final Justification:**

The rebuttal has adequately addressed my concerns. I think that extending the coverage of this work to autoregressive models increases the strength of this contribution. Thus, I will increase my score to clear Acceptance.

**Key Questions For Authors:**

Q1: Can the authors provide any empirical (preferably) or theoretical evidence that their approach is effective for paradigms beyond diffusion / flow-matching? In particular, for token-prediction image autoregressive models. Addressing this weakness will change my score to clear acceptance, unless I missed some issues spotted by other reviewers.

Q2: Will the authors consider extending HIFI-Gen with images generated by any SOTA model beyond diffusion / flow-matching?

I mention "Han et al. Infinity: Scaling Bitwise AutoRegressive Modeling for High-Resolution Image Synthesis" only as an example. Please do not feel bound to it in any way. Any example of this paradigm is sufficient to address my concerns. To be clear: my personal view is that GANs are no longer a thriving paradigm, so I do not particularly see much sense in extending in this direction.

**Limitations:**

Yes, although it misses the concerns I raised as weaknesses.

**Strengths And Weaknesses:**

Strengths:
- This is a genuinely novel, interesting approach. Shifting away from training on massive datasets to obtain detection capabilities is a key strength of DNA.
- The addressed problem is significant and worthy of study
- Extensive evaluation (with a major weakness, see below)
- Opensourcing the HIFI-Gen benchmark strengthens the contributions (however, the weakness below still applies)

Weaknesses:
- **All evaluated models are diffusion / flow-matching.** The generalisation to other SOTA paradigms is not tested. In particular, image autoregressive models (such as Infinity [1]) are not included.
- This weakness extends to  HIFI-Gen benchmark, limiting this contribution

I believe that the paper makes a novel, strong contribution that is undermined by the somewhat redundant and limiting model selection.
Regretfully, the model selection for both empirical evaluation of DNA and the HIFI-Gen benchmark **only encompasses diffusion / flow-matching models**, and those do not include other SOTA paradigms such as token prediction image autoregressive models.
Recommending rejection would be unfair, as the work is still valuable and its strengths lie in the novelty of the approach. However, I cannot currently give a higher score to this work.

[1] Han et al. Infinity: Scaling Bitwise AutoRegressive Modeling for High-Resolution Image Synthesis

---

> ### Author Rebuttal · Authors · 2026-03-31
>
> We sincerely thank the reviewer `nwaT` for **recognizing the novelty of our Discriminative Neural Anchors (DNA) framework and the value of our training-free approach**. We are especially grateful for your constructive suggestion regarding the inclusion of non-diffusion paradigms. We completely agree that this was a missing piece, and addressing it significantly strengthens both our evaluation and the HIFI-Gen benchmark.
>
> **W1 & Q1: Generalization to other SOTA paradigms (e.g., autoregressive models)**
>
> **A:** Thank you for your question. As suggested, we have extended our evaluation to include SOTA token-prediction image autoregressive models, explicitly incorporating Infinity, alongside other recent advanced paradigms like Nano Banana 2, Nano Banana Pro, and Della 3.
>
> A key strength of DNA, as you rightfully noted, is that it does not depend on external training. To demonstrate its robust generalization, we directly applied our established DNA framework to these entirely new paradigms **without any retraining, fine-tuning, or parameter adjustments**.
>
> As shown in the table below, DNA maintains exceptionally high detection accuracy across these diverse generative architectures, proving that its effectiveness is not isolated to diffusion or flow-matching models.
>
> | Model | Metric(↑)  | MoLD | DNA |
> | :--- | :--- | :--- |:--- |
> | Infinity | ACC/AP | 66.2/91.8 |**97.0**/**99.1** |
> | Nano Banana 2 | ACC/AP | 88.7/98.6 |**97.3**/**99.6** |
> | Nano Banana Pro |ACC/AP |89.0/97.6 | **96.8**/**98.7** |
> | Della 3 | ACC/AP | 50.0/48.9 |**88.3**/**94.3** |
>
> Thank you again for your insightful advice.
>
> **W2 & Q2: Extending the HIFI-Gen benchmark**
>
> **A:** Yes, absolutely. We have expanded the HIFI-Gen benchmark to include diverse high-resolution samples from Infinity, Nano Banana 2, Nano Banana Pro, and Della 3. Following the Gen-Image methodology, To maintain evaluation consistency, we utilized a standardized 'photo of a [class]' prompt structure. Due to the strict time constraints of the rebuttal period, we generated an initial set of 300 images per new model. We sincerely thank you again for your valuable suggestion.
>
>
> **Lastly, thank you so much for helping us improve the paper and for your constructive feedback! We are actively available until the end of this rebuttal period and look forward to hearing back from you.**

---

> > ### Author Rebuttal · Reviewer_nwaT · 2026-04-04
> >
> > I thank the Authors for the provided answers. My concerns have been adequately addressed. I think that extending the coverage of this work to autoregressive models increases the strength of this contribution. Thus, I will increase my score.

---

> > > ### Author Response · Authors · 2026-04-05
> > >
> > > Dear Reviewer nwaT,
> > >
> > > Thank you for your positive feedback! We are pleased that our response adequately addressed your concerns, especially regarding the extension of our work to autoregressive models. We are also very encouraged by the increased score. Thank you for your time and continued support!
> > >
> > > Sincerely,
> > >
> > > The Authors of Submission 8386

---

### Decision · Program_Chairs · 2026-04-30

**Decision:**

Accept (regular)

**Comment:**

This paper received 1 accept (after rebuttal), 2 weak accept and 1 weak reject. It presents a novel and practically meaningful approach to forgery detection by uncovering sparse discriminative neurons within pretrained models. The proposed DNA framework, along with the HIFI-Gen benchmark, demonstrates strong empirical performance across datasets, generators, and few-shot settings.

The main concerns include the strength of the ``intrinsic capability`` claim, the degree of novelty beyond standard components, and questions about adversarial robustness. The rebuttal substantially strengthened the paper by expanding evaluation to additional generative paradigms, improving reproducibility, and providing deeper empirical analysis.

A remaining concern is the unexpectedly strong robustness under white-box PGD attacks. While the authors provided additional checks and stronger attack settings, the robustness remains somewhat surprising and would benefit from more comprehensive evaluation. This issue, however, primarily affects the interpretation of robustness rather than the core contribution.

Overall, the strengths, particularly the empirical effectiveness, practical relevance, and improved evaluation, outweigh the remaining weaknesses. Thus, this meta review recommends weak accept. The paper will benefit from addressing the claim of the intrinsic capability and resolving the concerns about the PGD robustness (e.g., by testing AutoAttack instead).